# DNABERT-2: Efficient Foundation Model and Benchmark For Multi-Species Genomes

**Zhihan Zhou**[†]  **Yanrong Ji**[†]  **Weijian Li**[†]  **Pratik Dutta**[‡]  **Ramana V Davuluri**[‡]  **Han Liu**[†]

[†] Department of Computer Science, Northwestern University, Evanston, IL, USA

[‡] Department of Biomedical Informatics, Stony Brook University, Stony Brook, NY, USA

{zhihanzhou, yanrongji, weijianli}@u.northwestern.edu
pratik.dutta@stonybrook.edu, Ramana.Davuluri@stonybrookmedicine.edu
hanliu@northwestern.edu

## Abstract

Decoding the linguistic intricacies of the genome is a crucial problem in biology, and pre-trained foundational models such as DNABERT and Nucleotide Transformer have made significant strides in this area. Existing works have largely hinged on *k-mer*, fixed-length permutations of A, T, C, and G, as the *token* of the genome language due to its simplicity. However, we argue that the computation and sample inefficiencies introduced by k-mer tokenization are primary obstacles in developing large genome foundational models. We provide conceptual and empirical insights into genome tokenization, building on which we propose to replace k-mer tokenization with Byte Pair Encoding (BPE), a statistics-based data compression algorithm that constructs *tokens* by iteratively merging the most frequent co-occurring genome segment in the corpus. We demonstrate that BPE not only overcomes the limitations of k-mer tokenization but also benefits from the computational efficiency of non-overlapping tokenization. Based on these insights, we introduce DNABERT-2, a refined genome foundation model that adapts an efficient tokenizer and employs multiple strategies to overcome input length constraints, reduce time and memory expenditure, and enhance model capability. Furthermore, we identify the absence of a comprehensive and standardized benchmark for genome understanding as another significant impediment to fair comparative analysis. In response, we propose the Genome Understanding Evaluation, a comprehensive multi-species genome classification dataset that amalgamates 36 distinct datasets across 9 tasks, with input lengths ranging from 70 to 10000. Through comprehensive experiments on the GUE benchmark, we demonstrate that DNABERT-2 achieves comparable performance to the state-of-the-art model with 21× fewer parameters and approximately 92× less GPU time [1] in pre-training. Compared to DNABERT, while being 3× more efficient, DNABERT-2 outperforms it on 23 out of 28 datasets, with an average improvement of 6 absolute scores on GUE. The code, data, and pre-trained model are available at https://github.com/MAGICS-LAB/DNABERT_2.

## 1 Introduction

Transformer-based foundation models (Bommasani et al., 2022; Kenton & Toutanova, 2019; OpenAI, 2023) have witnessed significant progress in recent years, particularly exemplified by the advent of groundbreaking language models like ChatGPT (Ouyang et al., 2022; OpenAI, 2023). In parallel, the significance of foundation models has also been increasingly appreciated in the genomics field, as they represent the understanding of genome sequences via numerical embeddings that are directly applicable to various genome analysis tasks. These models can capture complex relationships and dependencies in DNA sequences, opening new avenues for understanding transcriptional regulation (Li et al., 2023), non-coding genetic variants associated with human diseases and traits (Rozowsky et al., 2023), and the functional effects of regulatory elements (Smith et al., 2023). Recent advancements in genome language modeling have demonstrated their superiority in a range of downstream applications, including promoter prediction (Le et al., 2022; Zhang et al., 2022), gene expression

---

[1]About 14 days on 8 NVIDIA RTX 2080Ti V.S. 28 days on 128 NVIDIA A100. Estimated with the **Method 2: GPU Time** introduced by OpenAI in https://openai.com/research/ai-and-compute.

prediction (Avsec et al., 2021), DNA methylation prediction (Jin et al., 2022), chromatin state analysis (Lee et al., 2022), promoter-enhancer interaction prediction (Chen et al., 2022; Ni et al., 2022), TF-DNA binding prediction (Wang et al., 2022), variant effect prediction (Rozowsky et al., 2023), gene network prediction (Theodoris et al., 2023) and more. These models provide researchers with powerful tools to understand the functional importance of different genomics elements and unravel key biological processes and mechanisms.

In this context, Ji et al. (2021) developed DNABERT, an initial foundation model (FM), to unravel the human genome from a language perspective. Despite being widely applied in the community, several technical limitations still present at the time with the original DNABERT implementation, limiting its full potential. First, although proven to be generalizable to other organisms, the pretraining was solely done on the human reference genome, omitting the sequence conservation and diversity across species. Second, k-mer tokenization resulted in information leakage and overall poor computational efficiency during pre-training, which hampers its scalability. Lastly, the simplistic DNABERT-XL solution—intended to bypass the restriction of 512 input sequences imposed by the learned positional embedding (Kenton & Toutanova, 2019)—fell short in handling long input sequences, both in efficiency and effectiveness. These limitations underlined the need for further advancements in the domain of DNA language models.

Recently, Dalla-Torre et al. (2023) introduced Nucleotide Transformers (NT), a series of genome foundation models scaling from $500M$ to $2500M$ parameters. NT alleviated the first two limitations of DNABERT by pre-training on a large collection of genomes from 850 species and replacing overlapping k-mer tokenization with a non-overlapping version, substantially reducing tokenized sequence length. Despite this, a hard input length limitation still exist, while, as we will discuss in Sec. 2, non-overlapping k-mer tokenization also suffered from poor sample efficiency as it complicates the model's task of aligning significantly distinct representations of near-identical inputs.

In view of the aforementioned limitations, we introduce DNABERT-2, a multi-species genome foundation model that replaces k-mer tokenization with Byte Pair Encoding (BPE) (Sennrich et al., 2016), a data compression algorithm that has been widely used by large language models. We show that BPE effectively addresses the known issues of k-mer tokenization while maintaining the computational efficiency of non-overlapping tokenization. Moreover, DNABERT-2 overcomes the limitation of DNABERT by replacing learned positional embeddings with Attention with Linear Biases (ALiBi) (Press et al., 2021) to get rid of the input length limitation, incorporating Flash Attention (Dao et al., 2022) to increase computational efficiency, and adjusting model architecture to increase model capability. As a result of the efficient tokenizer and advanced model architecture, DNABERT-2 achieves comparable performance to the state-of-the-art model with approximately 92× less computational cost and 21× fewer parameters, identifying its computation- and sample-efficiency and enabling efficient fine-tuning on most consumer GPUs.

Meanwhile, despite progress in genome foundational models, the absence of carefully curated benchmarks has posed a significant challenge. Owing to the unstandardized pre-processing pipeline of genome sequences, it is unjust to directly compare model performances with results reported in previous papers, even when the data originate from the same source. Moreover, many genome understanding evaluation datasets used in existing works (Dalla-Torre et al., 2023) are either too trivial or too challenging, leading to similar scores for most models and failing to accurately reflect different models' capabilities. The scarcity of high-quality benchmark datasets hampers evaluating and comparing different models and further hinders the development of novel techniques. To this end, we introduce Genome Understanding Evaluation (GUE & GUE$^{+}$), a standardized and comprehensive multi-species benchmark containing 36 datasets across 9 important genome analysis tasks on genomes of 4 species with input lengths ranging from 70 to 10000. All the datasets are elaborately calibrated with a series of strategies to ensure they are suitable for reflecting the capability level of existing genome foundation models.

Our main contributions can be therefore summarized as follows: 1) We identify key obstacles in genome tokenization and provide deep insights, presenting a simple yet effective solution that balances the efficiency and effectiveness of genome foundation models; 2) We introduce DNABERT-2, an efficient pre-trained foundation model for multi-species genome that delivers performance on par with the state-of-the-art model while being 21× smaller and utilizes approximately 92× less GPU time; 3) We introduce Genome Understanding Evaluation (GUE), a standardized, comprehensive,

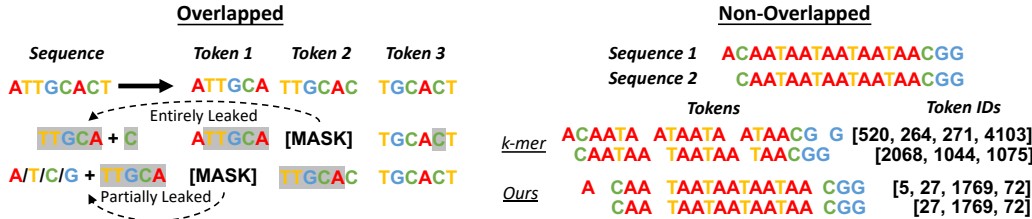

Figure 1: Illustration of the drawbacks of k-mer tokenization. In the overlapping setting, information about a masked token is leaked by its adjacent tokens, while in the non-overlapping setting, adding/deleting one nucleotide base leads to a dramatic change in the tokenized sequence.

and well-calibrated multi-species genome classification benchmark including 9 tasks and 36 datasets to facilitate research in genome foundation model.

## 2 BACKGROUND

Tokenization serves as a critical initial step in language modeling, significantly impacting the efficiency and effectiveness of the model. DNA sequences consist of 4 unique nucleotide bases: A, T, C, and G. A majority of genome language models (Ji et al., 2021; Dalla-Torre et al., 2023) utilize the k-mer tokenization technique, in which each contiguous $k$-length genome segment is considered as a token. During tokenization, a sliding window with window size $k$ and stride $t$ is employed to convert the original genome sequence into a series of k-mers. Here, the stride $t$ is either set as $1$ or $k$, while the first one represents the overlapping version of k-mer tokenization and the other one represents the non-overlapping version. Figure 1 presents examples of overlapping (left) and non-overlapping (right) k-mer tokenizations. Despite its wide application, we argue that both versions of the k-mer tokenization are less optimal.

Overlapping k-mers tokenization ensures adjacent tokens always overlap by $k-1$ characters, resulting in significant information leakage in masked language modeling. As depicted in Figure 1, a masked token is entirely leaked when adjacent tokens from both sides are not masked, and it is partially leaked when adjacent tokens from only one side are present. Generally, in the overlapping $k$-mer tokenization setting, let $l$ and $r$ denote the distances between a masked token [M] and its closest unmasked adjacent token on the left and right sides, the number of possible options of [M] is $4^{\min(l,r,k,\max(0,l+r-k))}$. In other words, to prevent the entire leakage of a masked token, at least $k-1$ tokens on its left and right sides in total must be masked, which explains why Ji et al. (2021) opt to mask a continuous span of $k$ tokens. Furthermore, to guarantee no leakage of a masked token, at least $k$ tokens on both sides must be masked. Nevertheless, information leakage is still inevitable for the leftmost and rightmost $k-1$ masked tokens. Ideally, in masked language modeling, a model is required to select the best option from the *entire* vocabulary, enabling it to differentiate and evaluate among a large number of options. However, if the search space is undesirably reduced due to information leakage, the model only needs to differentiate between a limited number of options. Consequently, this results in poor sample efficiency, as the model may not be sufficiently challenged to learn the underlying patterns in the data. Also, the tokenized sequence for an input of length $L$ consists of $L-k+1$ tokens, each with a length of $k$. This results in a tokenized sequence with considerable redundancy and a length nearly equivalent to the original sequence, leading to low computation efficiency considering the quadratic computation complexity of Transformer-based (Vaswani et al., 2017) models. This becomes particularly problematic when attempting to scale up the model. Therefore, Dalla-Torre et al. (2023) proposed the non-overlapping k-mer tokenization.

Non-overlapping k-mer tokenization, despite its advantage of reducing sequence length by a factor of $k$, is plagued by a notable issue of sample inefficiency. Figure 1 graphically illustrates this problem. Considering a scenario when the context window is reduced by $1$, the model input is then switched from **Sequence 1** to **Sequence 2**. In theory, this should involve a minor adjustment in tokenized output. However, with the non-overlapping k-mer tokenizer, this minor shift instigates a dramatic alteration in the tokenized output. Despite the two sequences originating from the same genomic segment, their tokenized representations bear little resemblance. This inconsistent behavior introduces

unnecessary hurdles for the model during training, as it poses unnecessary difficulty for the model to align distinct representations of identical or near-identical inputs. Consequently, the inefficiency in learning from the data could impede the overall model performance. The implications of these observations advocate for a re-evaluation of tokenization strategies for the genome language, with a focus on strategies that ensure robust and efficient representation.

To address the aforementioned issues, we propose to adapt SentencePiece (Kudo & Richardson, 2018), a subword tokenization framework widely used in natural language processing, to replace k-mer tokenization for genome sequences. We employ Byte-Pair Encoding (BPE) (Sennrich et al., 2016) to iteratively merge frequent pairs of nucleotides and genome segments, forming a vocabulary of variable-length tokens that effectively represent the entire genome dataset. Despite its conceptual simplicity, this method is well-suited for genome foundation models. First, it not only prevents information leakage but also significantly reduces the sequence length by approximately 5 times (detailed statistics are presented in Sec 3.1), substantially improving computational efficiency. Moreover, its robust tokenization result is beneficial for sample efficiency since it allows the model to focus on understanding the genome language semantics without being distracted by the distinct representations of the same input. Furthermore, unlike k-mer tokenization, BPE doesn't always produce tokens of length $k$. Consequently, when a token containing an unspecified number of nucleotides is masked, the model is challenged to predict both the number of nucleotides and the particular nucleotides themselves. This naturally transforms the masked language modeling objective into a T5-style (Raffel et al., 2020) "replace spans of text" objective, which has been demonstrated to be more effective than standard masked language modeling in various scenarios.

## 3 METHOD

In this section, we provide empirical analysis on the BPE tokenizer for genome language (§ 3.1) and describe the model architecture (§ 3.2) and implementation details (§ A.2) of DNABERT-2.

### 3.1 TOKENIZER

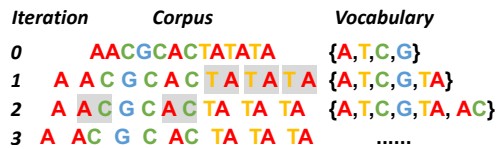

| Iteration | Corpus | Vocabulary |
|---|---|---|
| 0 | AACGCACTATATA | {A,T,C,G} |
| 1 | A A C G C A C T A T A T A | {A,T,C,G,TA} |
| 2 | A A C G C A C TA TA TA | {A,T,C,G,TA, AC} |
| 3 | A AC G C AC TA TA TA | ...... |

Figure 2: Illustration of the BPE vocabulary constructions.

DNABERT-2 adapts SentencePiece (Kudo & Richardson, 2018) with Byte Pair Encoding (BPE) (Sennrich et al., 2016) to perform tokenization for DNA sequences. SentencePiece is a language-agnostic tokenizer that considers each input as a raw stream without assuming any pre-tokenization, which matches greatly with genome sequences where the definitions of *word* and *sentence* do not exist. BPE is a compression algorithm that has been widely used in the area of natural language processing as a word segmentation strategy. It learns a fixed-sized vocabulary of variable-length tokens based on the co-occurrence frequency of the characters. Figure 2 illustrates the process of constructing a vocabulary from a given corpus with BPE. First, we initialize the vocabulary with all unique characters in the corpus. Then, in each iteration, we view the most frequent character segment (e.g., TA at iteration 1) as a new *word*, add it to the vocabulary, and update the corpus by replacing all the same segments with this new word. The iteration continues till we achieve the desired number of words in the vocabulary. Thus, the target vocabulary size plays a crucial role.

Due to the significant difference between natural language and DNA sequence, vocabulary sizes that are commonly used in the NLP area (Kenton & Toutanova, 2019; Vaswani et al., 2017; Raffel et al., 2020; OpenAI, 2023) may not be appropriate for genome sequences. To determine the most suitable vocabulary size, we constructed 8 vocabularies with target sizes ranging from $2^8$ to $2^{15}$ on the multi-species genomes (see Sec. 4.1) to empirically evaluate the impact of varying vocabulary sizes. As indicated in Figure 3a, larger vocabularies tend to encompass more lengthy tokens, which enables the tokenizer to represent the same input sequence with fewer tokens. Shorter tokenized sequences consequently reduce the computational cost (See Figure 3b), as the computational complexity of Transformers is quadratic in relation to the input sequence length. Therefore, from the computation efficiency perspective, a larger vocabulary size is favorable.

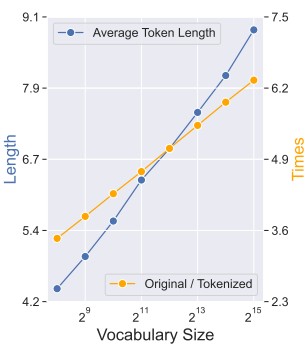 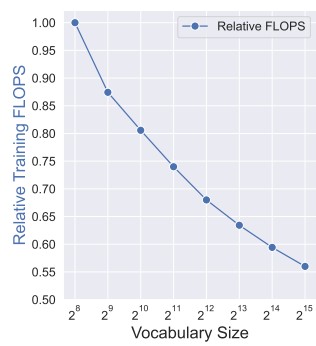 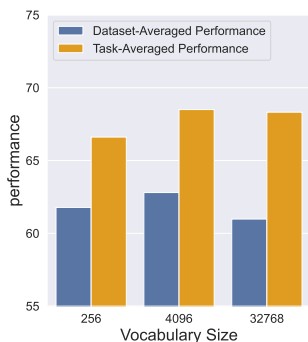

(a) Average token length and the length ratio of original sequence v.s. tokenized sequence.

(b) Training FLOPs on 500-length sequences compared to model with $2^8$ vocabulary.

(c) Model performance averaged over each tasks (macro) and individual dataset (micro).

Figure 3: This figure presents the average token length, average sequence length reduced after tokenization, and model performance on the GUE benchmark with different vocabulary sizes.

However, a larger vocabulary leads to more sparse updates to the embedding layer, given that each token would be used less frequently, which might compromise the model's performance. We empirically analyzed this issue by pre-training three different DNABERT-2 variants with vocabulary sizes of $2^8$, $2^{12}$, and $2^{15}$ on the multi-species genome dataset with a batch size of 2048 for 150000 steps and evaluating them on the GUE benchmark (see Sec. 4.2). Figure 3c displays the performance of each variant, where the model performance is measured by the dataset- and task-average scores. As depicted in the figure, unlike computational efficiency, the model's performance does not consistently improve as the vocabulary size increases. Therefore, we selected a vocabulary size of $2^{12} = 4096$ for training the final DNABERT-2 model, as it best balances model performance with computational efficiency among the candidates.

## 3.2 MODEL

DNABERT-2 adapts the Transformer Encoder architecture. To address the limitations of existing models, we incorporate a series of recent advances in deep learning to increase the model's efficiency and capability, including: 1) replacing learned positional embeddings with the Attention with Linear Biases (ALiBi) (Press et al., 2021) to overcome the input length limitation; 2) utilizing FlashAttention (Dao et al., 2022) and Low Precision Layer Normalization to increase computation and memory efficiency; 3) employing the Low-Rank Adaptation (LoRA) (Hu et al., 2021) in the fine-tuning stage (if necessary) for parameter-efficient training.

**Attention with Linear Biases.** Due to the permutation-invariant nature of the attention mechanism, explicit positional information is required in attention-based models. Existing solutions such as Sinusoidal (Vaswani et al., 2017), learned (Kenton & Toutanova, 2019), and Rotary (Su et al., 2021) positional embedding methods either suffer from input length restriction or poor *extrapolation* capability when applied to sequences longer than training data. Attention with Linear Biases (ALiBi) provides an efficient yet effective solution. Instead of adding position embeddings to the input, ALiBi adds a fixed set of static, non-learned biases to each attention calculation to incorporate positional information into attention scores. Specifically, let $\mathbf{q}_i$ define the $i$-$th$ query in the input sequence of length $L$ and $\mathbf{K}$ defines the key matrix, the attention score of query $i$ is calculated as: $\mathrm{softmax}(\mathbf{q}_i\mathbf{K} + m * [-(i-1), ..., -2, -1, 0, -1, -2, ..., -(L-1-i)])$, where $m$ is a fixed head-specific constant. ALiBi used a geometric sequence (i.e., $\frac{1}{2^1}, \frac{1}{2^2}, ..., \frac{1}{2^n}$) of different $m$ to each attention head. Intuitively, ALiBi increasingly penalizes attention scores between key-query pairs as their distances increase, and $m$ determines the penalty rate. Replacing learned position embedding with ALiBi allows DNABERT-2 to effectively handle arbitrarily long sequences during fine-tuning and inference despite being pre-trained on relatively short sequences.

**Flash Attention.** Flash attention is an IO-aware algorithm that implements the exact standard attention calculation in a more time- and memory-efficient way. It identifies a main bottleneck of standard attention implementation as the lack of taking the number of reads and writes to fast GPU on-chip SRAM and relatively slow GPU high bandwidth memory (HBM) into account. To avoid reading and writing to the slow HBM, it splits Key/Query/Value matrices into blocks and incrementally performs softmax over the entire input. It also proposes to recompute large intermediate results like attention scores in backward pass to trade extra computation for fewer IO with HBM, which empirically leads to less computational time without sacrificing model performance.

**Low-Rank Adaptation (LoRA).** Fine-tuning all the parameters of a model becomes increasingly expensive as the pre-trained model becomes much larger. Thus, we adopt LoRA, a parameter-efficient fine-tuning method that significantly reduces the computation and memory costs with ignorable performance sacrifice. Let $W_0, W_1 \in \mathbb{R}^{m \times n}$ define the same weight matrix before and after task-specific fine-tuning, and we have $W_1 = W_0 + \Delta W$, where $\Delta W \in \mathbb{R}^{m \times n}$ represents the change of each weight element during the fine-tuning. In ordinary fine-tuning, we independently update each weight based on its corresponding gradient, while in LoRA, we represent $\Delta W$ with a low-rank decomposition $\Delta W = BA$, where $B \in \mathbb{R}^{m \times r}$, $A \in \mathbb{R}^{r \times n}$, and $r \ll m, r \ll n$. Modeling $\Delta W$ with low-rank decomposition reduces the number of trainable parameters from $m \times n$ to $r \times (m + n)$, leading to significant improvement in training time and memory usage.

Besides, we replace the Relu activation function with GEGLU (Shazeer, 2020), a variant of GLU (Dauphin et al., 2017) that has been shown to improve the performance of Transformer models. The GEGLU function is calculated as $\texttt{GEGLU}(x, W, V, b, c) = \texttt{GELU}(xW + b) \otimes (xV + c)$, where $x$ is the function input, $W$ and $V$ are learnable weights, and $b$ and $c$ are learnable biases. The GELU function is defined as $\texttt{GELU}(x) = x\Phi(x)$, where $\Phi(x)$ is the cumulative distribution function (CDF) of the standard normal distribution.

## 4 DATA

In order to facilitate further research on large-scale genome foundational models, we have collated and made available multi-species genome datasets for both pre-training of models (Sec. 4.1) and benchmarking (Sec. 4.2).

### 4.1 PRE-TRAIN: HUMAN AND MULTI-SPECIES GENOME

To investigate the impact of species diversity on genome foundation models, we compile and made publicly available two datasets for foundation model pre-training: the human genome and the multi-species genome. The human genome dataset is the one used in DNABERT (Ji et al., 2021), which comprises 2.75B nucleotide bases. The multi-species genome dataset encompasses genomes from 135 species, spread across 6 categories. In total, this dataset includes 32.49B nucleotide bases, nearly 12 times the volume of the human genome dataset. We exclude all sequences with `N` and retain only sequences that consist of `A`, `T`, `C`, and `G`. Detailed statistics are presented in Table 11.

### 4.2 BENCKMARK: GENOME UNDERSTANDING EVALUATION (GUE & GUE$^+$)

We introduce a large benchmark for genome foundation models. Due to the input length limits of existing genome foundation models, we split the benchmark into two parts: GUE and GUE$^+$. GUE (See Table 1) includes 7 genome sequence classification problems with 28 datasets with input lengths ranging from 70 to 1000. GUE$^+$ (See Table 2), on the other hand, focuses on relatively longer input sequences, rating from 5000 to 10000. To evaluate the multi-species transferability in genome understanding of each model, this benchmarks contains tasks for various species, including human, fungi, virus, and yeast. We explicitly define evaluation metrics for each task and split each dataset into training, validation, and test data for a fair comparison across different models.

To calibrate the benchmark's difficulty level and better illuminate each model's capabilities, we carefully selected datasets that are neither too simple nor overly challenging for current models. For example, when the Nucleotide Transformer variants (Dalla-Torre et al., 2023) were tested on the *Splice Site Prediction* dataset, all variants achieved an accuracy between 97% and 98%. Similar

| Species | Task | Num. Datasets | Num. Classes | Sequence Length |
|---|---|---|---|---|
| **Human** | Core Promoter Detection | 3 | 2 | 70 |
| | Transcription Factor Prediction | 5 | 2 | 100 |
| | Promoter Detection | 3 | 2 | 300 |
| | Splice Site Detection | 1 | 3 | 400 |
| **Mouse** | Transcription Factor Prediction | 5 | 2 | 100 |
| **Yeast** | Epigenetic Marks Prediction | 10 | 2 | 500 |
| **Virus** | Covid Variant Classification | 1 | 9 | 1000 |

Table 1: Summarization of the Genome Understanding Evaluation (GUE) benchmark.

| Species | Task | Num. Datasets | Num. Classes | Sequence Length |
|---|---|---|---|---|
| **Human** | Enhancer Promoter Interaction | 6 | 2 | 5000 |
| **Fungi** | Species Classification | 1 | 25 | 5000 |
| **Virus** | Species Classification | 1 | 20 | 10000 |

Table 2: Summarization of the Genome Understanding Evaluation Plus (GUE$^+$) benchmark.

outcomes were observed in tasks like *Promoter Prediction* and *Enhancer Prediction*. These high scores might suggest these variants perform similarly, but as our experiments in Section 5 show, they vary significantly on more discerning datasets.

The construction of the benchmark starts with the aggregation of various biologically important genome analysis datasets, followed by the assessment of existing models such as DNABERT (Ji et al., 2021) and Nucleotide Transformer (Dalla-Torre et al., 2023) on these datasets. Datasets where the majority of models yielded moderate (e.g., F1-scores between 0.3 and 0.8) and distinguishable performance scores were retained. On the other hand, datasets that did not meet these criteria underwent a restructuring process involving various strategies such as class balancing, adversarial sample inclusion, and reduction of training sample volume, among others. After several iterations of this process, we ultimately arrived at 36 representative datasets of moderate difficulty. Due to space limits, we present the detailed data processing and statistics of each dataset in Sec. B.2.

## 5 EXPERIMENTS

We evaluate DNABERT-2 using the proposed benchmark to thoroughly investigate its versatility and robustness across a variety of tasks involving multi-species genomes.

### 5.1 BASELINE

We compare DNABERT-2 with two state-of-the-art genome foundation models: DNABERT (Ji et al., 2021) and Nucleotide Transformer (Dalla-Torre et al., 2023).

**DNABERT** was the first pre-trained foundational model for genome sequences, trained on human genome sequences. It has four variants, namely *DNABERT (3-mer)*, *DNABERT (4-mer)*, *DNABERT (5-mer)*, and *DNABERT (6-mer)*, which utilize overlapping 3/4/5/6-kmer tokenization respectively. While DNABERT employs the same architecture as `BERT-base`, it has a different vocabulary size, which is dependent on the chosen $k$-mer.

**Nucleotide Transformer (NT)** scales up the data and model size to achieve state-of-the-art performance in 27 DNA analysis tasks. It also has 4 variants: *NT-500M-human*, *NT-500M-1000g*, *NT-2500M-1000g*, and *NT-2500M-multi*, where *human*, *1000g*, and *multi* respectively refers to the GRCh38/hg38 human reference genome, 3202 high-coverage human genomes from the 1000 Genome project (Byrska-Bishop et al., 2021), and genome from 850 different species.

It is important to note that NT models are 6 to 29 times larger than DNABERT, which precludes standard model fine-tuning on consumer GPUs. Therefore, we perform standard fine-tuning for

| Model | Params. ↓ | FLOPs ↓ | Trn. Tokens | Num. Top-2 ↑ | Ave. Scores ↑ |
|---|---|---|---|---|---|
| **DNABERT (3-mer)** | 86M | 3.27 | 122B | 2 ‖ 0 | 61.62 |
| **DNABERT (4-mer)** | 86M | 3.26 | 122B | 0 ‖ 1 | 61.14 |
| **DNABERT (5-mer)** | 87M | 3.26 | 122B | 0 ‖ 1 | 60.05 |
| **DNABERT (6-mer)** | 89M | 3.25 | 122B | 0 ‖ 1 | 60.51 |
| **NT-500M-human** | 480M | 3.19 | 50B | 0 ‖ 0 | 55.43 |
| **NT-500M-1000g** | 480M | 3.19 | 50B | 0 ‖ 1 | 58.23 |
| **NT-2500M-1000g** | 2537M | 19.44 | 300B | 0 ‖ 1 | 61.41 |
| **NT-2500M-multi** | 2537M | 19.44 | 300B | 7 ‖ 9 | 66.93 |
| **DNABERT-2** | 117M | 1.00 | 262B | 8 ‖ 4 | 66.80 |
| **DNABERT-2♦** | 117M | 1.00 | 263B | **11** ‖ **10** | **67.77** |

Table 3: The statistics and performance of each model. The five columns represent the number of model parameters, relative FLOPs compared to DNABERT-2, the number of tokens used in pre-training, and the number of being top-2 among all the models (1st ‖ 2nd) and the average evaluation scores on the 28 datasets of the GUE benchmark. ♦: perform further masked language modeling pre-training on the training sets of the GUE benchmark.

DNABERT and DNABERT-2, while adapting the Low-Rank Adaptation (LoRA) technique for fine-tuning the Nucleotide Transformer to enhance efficiency. For a fair comparison, we conducted preliminary experiments to confirm that our implementation of NT achieves comparable results to those reported in their original paper (Dalla-Torre et al., 2023) (see Appendix A.4 for more details).

## 5.2 SETUP AND METRIC

We evaluate the models from two perspectives: computational efficiency and performance on downstream tasks. To measure each model's computational cost, we consider the number of model parameters and the relative Floating Point Operations (FLOPs)—which is the total number of multiplication and addition operations during a forward pass—compared to DNABERT-2. We evaluate FLOPs on genome sequences with a length of 500, a commonly used setup in genome analysis. To measure model performance, we utilize F1-Score and Matthews Correlation Coefficient (MCC). We use different metrics for different tasks, following conventional practices (refer to Table 12 for details). Table 3 presents the overall performance of each model on the GUE benchmark. It provides the average score of each model and the number of times it ranks in the top two among all models. The average results across all tasks are reported in Table 4, while task-specific results are presented in Table 6 due to space constraints. We also include statistics on the number of tokens each model processed during its pre-training phase, providing insight into the effects of training steps on model performance. For each model, we keep most of the hyperparameters (e.g., learning rate, batch size, weight decay, etc.) constant across all datasets, adjusting only the maximum sequence length and the number of training steps according to the specific dataset. Hyperparameter tuning tailored to each dataset is left for future work. Throughout the training process, we validate the model every 200 steps, save the model that yields the smallest loss on the validation set, and report its evaluation results on the test set. We train each model using three different random seeds and report the average results.

**Further Pre-Training.** We also investigate the impact of additional in-domain pre-training on DNA language models. We combine the training sets of the 28 GUE datasets and further pre-train DNABERT-2 on this combined set. Following Sun et al. (2020), we train the model with a batch size of 32, a maximum sequence length of 128, and a learning rate of $5e-5$ for 100,000 steps. This results in 0.41B training tokens, which only constitute 0.08% of the tokens processed during the entire training process of DNABERT-2.

## 5.3 RESULTS ON GUE

Table 3 outlines the statistics and aggregate performance of the models. As indicated in the table, despite being 21× smaller and requiring 19× fewer FLOPs, DNABERT-2 delivers a performance comparable to the state-of-the-art model while significantly surpassing other baselines. When DNABERT-2 undergoes additional pre-training on the GUE benchmark, which requires negligible computational overhead, it delivers the highest average performance and consistently ranks in the

| Species | Yeast | Mouse | Virus | Human | | | |
|---|---|---|---|---|---|---|---|
| Task | EMP | TF-M | CVC | TF-H | PD | CPD | SSP |
| DNABERT (3-mer) | 49.54 | 57.73 | 62.23 | 64.43 | 84.63 | **72.96** | 84.14 |
| DNABERT (4-mer) | 48.59 | 59.58 | 59.87 | 64.41 | 82.99 | 71.10 | 84.05 |
| DNABERT (5-mer) | 48.62 | 54.85 | 63.64 | 50.46 | 84.04 | 72.03 | 84.02 |
| DNABERT (6-mer) | 49.10 | 56.43 | 55.50 | 64.17 | 81.70 | 71.81 | 84.07 |
| NT-500M-human | 45.35 | 45.24 | 57.13 | 50.82 | 85.51 | 66.54 | 79.71 |
| NT-500M-1000g | 47.68 | 49.31 | 52.06 | 58.92 | 86.58 | 69.13 | 80.97 |
| NT-2500M-1000g | 50.86 | 56.82 | 66.73 | 61.99 | 86.61 | 68.17 | 85.78 |
| NT-2500M-multi | 58.06 | 67.01 | **73.04** | 63.32 | **88.14** | 71.62 | **89.36** |
| DNABERT-2 | 55.98 | 67.99 | 71.02 | **70.10** | 84.21 | 70.52 | 84.99 |
| DNABERT-2♦ | **58.83** | **71.21** | 68.49 | 66.84 | 83.81 | 71.07 | 85.93 |

Table 4: The models' averaged performance on the 7 tasks in the GUE benchmark, including Epigenetic Marks Prediction (EMP), Transcription Factor Prediction on the Human genome and the Mouse genome (TF-H and TF-M), Covid Variants Classification (CVC), Promoter Detection (PD), Core Promoter Detection (CPD), and Splice Site Prediction (SSP).

top two across the 28 tasks of the GUE benchmark. These results showcase the model's remarkable efficiency and effectiveness.

Despite having 30% more parameters than DNABERT, DNABERT-2 requires only one-third the number of FLOPs. This indicates the superiority of the Byte Pair Encoding (BPE)-based tokenization method over overlapping k-mer tokenization in terms of modeling efficiency. Armed with the new tokenization method and the Attention with Linear Biases (ALiBi) module, DNABERT-2 can effectively process long genome sequences arbitrarily, demonstrating enhanced efficiency. This improvement becomes even more significant as the length of the input sequence increases. Moreover, DNABERT-2 consistently outperforms DNABERT by a large margin, indicating the effectiveness of multi-species pre-training and new model architecture.

Although DNABERT-2 is 5 times smaller, it surpasses NT-500M while using fewer FLOPs. This underscores the importance of providing the model with *adequate* data, particularly when the model size is scaled up, and further highlights the inefficiency of overlapping k-mer tokenization. The comparison between DNABERT and NT-2500M-1000g exposes the sample inefficiency of non-overlapping k-mer tokenization. Despite being trained on 2.5 times more tokens, NT-2500M-1000g achieves a performance similar to that of DNABERT.

The averaged results for each task are displayed in Table 4. DNABERT-2 and NT-2500M-multi consistently achieve top-tier performance across most tasks. Their dominance over other baselines is particularly notable in non-human genome analysis tasks, demonstrating the effectiveness of multi-species genomes pre-training. Furthermore, models trained on multi-species genomes also show strong performance on human genome analysis tasks, proving their ability to develop a comprehensive understanding of multi-species genomes without compromising their grasp of the human genome. We also observe that additional pre-training does not uniformly enhance all tasks, indicating that task-specific further pre-training might be necessary when addressing a certain downstream task.

Additionally, DNABERT variants achieve optimal performance in the Core Promoter Detection task, where inputs are sequences of length 70. However, their performance diminishes in the similar task of Promoter Detection, where the input length increases to 300. These results highlight a common challenge associated with non-overlapping k-mer tokenization and BPE-based tokenization: the capacity to identify subtle signals from limited input. Although inefficient, the overlapping k-mer tokenization adopted by DNABERT retains most of the information in the original sequences. In contrast, the sequence length is significantly reduced (*i.e.,* from 70 to 15) with non-overlapping k-mer and BPE tokenization, which might limit the retained information and hinder informed decision-making. This identifies a critical area for future exploration in DNA language models.

| Task | SC (Fungi) | SC (Virus) | EPI (Human) | | | | | |
|------|------------|------------|---|---|---|---|---|---|
| **Dataset** | | | **0** | **1** | **2** | **3** | **4** | **5** |
| **DNABERT (6-mer)** | 89.29 | 44.51 | - | - | - | - | - | - |
| **NT-2500M-multi** | 92.85 | 45.00 | 61.91 | 72.15 | 73.13 | 79.49 | 86.48 | 68.64 |
| **DNABERT-2** | **93.04** | **48.50** | **76.21** | **79.19** | **83.50** | **86.71** | **92.90** | **73.70** |

Table 5: The models' averaged performance on the 3 tasks in the GUE$^+$ benchmark, including 6 Enhance Promoter Interaction (EPI) datasets and two Species Classification (SC) datasets on virus and fungi. The **-** means we are not able to fit the model on the dataset after trying multiple sets of hyperparameters.

## 5.4 RESULTS ON GUE$^+$

In this section, we further compare DNABERT-2 with DNABERT and Nucleotide Transformers on the GUE$^+$ benchmark. The input size of datasets in GUE$^+$ ranges from 5000 to 10000. For DNABERT, due to its 512 input limitation, we follow its origin setting by splitting the input into 512bp-len pieces, generating embedding independently, and feeding the averaged embedding to the output layer. Table 5 shows DNABERT, Nucleotide Transformer, and DNABERT-2's performance on the GUE$^+$ benchmark. As shown in the table, DNABERT-2 consistently outperforms the baselines on all the datasets, showing its good capability in handling long DNA sequences. Notably, despite DNABERT-2 is pre-trained purely on 700bp-len sequences, it performs well even on 10000-bp sequences with a few epochs of fine-tuning, indicating DNABERT-2's extrapolation capability offered by ALiBi.

## 6 CONCLUSION

In this paper, we introduce DNABERT-2, an efficient foundational model pre-trained on multi-species genomes. We identify the computational and sample inefficiencies of the existing k-mer tokenization method and propose the adaptation of Byte Pair Encoding (BPE) for DNA language modeling. We provide insightful and comprehensive empirical analyses, building DNABERT-2 based on these findings. Moreover, we integrate several techniques such as Attention with Linear Biases (ALiBi) and Low-Rank Adaptation (LoRA) to address the limitations of current DNA language models. We also compile and introduce the Genome Understanding Evaluation, a benchmark for multi-species genome analysis comprising 9 tasks and 36 datasets with well-defined data split, evaluation metrics, and elaborately calibrated difficulty. Empirical analysis on the GUE benchmark shows that DNABERT-2 achieves comparable performance as the SOTA models while being much more efficient. For future work, we are interested in effective modeling strategies for short and extra-long genome sequences with novel architectures (Hu et al., 2023; 2024) and the introduction of training targets and data processing/augmentation methods that leverage the unique double-strand structure of DNA.

## ACKNOWLEDGE

This work is supported by NIH R01LM01372201.

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

# A EXPERIMENTS

## A.1 ALL EXPERIMENT RESULTS

| | Epigenetic Marks Prediction | | | | | |
| --- | --- | --- | --- | --- | --- | --- |
| | H3 | H3K14ac | H3K36me3 | H3K4me1 | H3K4me2 | H3K4me3 |
| **DNABERT (3-mer)** | 74.15 | 42.07 | 48.49 | 42.95 | 31.34 | 28.92 |
| **DNABERT (4-mer)** | 73.03 | 41.88 | 48.03 | 41.06 | 30.66 | 25.31 |
| **DNABERT (5-mer)** | 73.40 | 40.68 | 48.29 | 40.65 | 30.67 | 27.10 |
| **DNABERT (6-mer)** | 73.10 | 40.06 | 47.25 | 41.44 | 32.27 | 27.81 |
| **NT-500M-human** | 69.67 | 33.55 | 44.14 | 37.15 | 30.87 | 24.06 |
| **NT-500M-1000g** | 72.52 | 39.37 | 45.58 | 40.45 | 31.05 | 26.16 |
| **NT-2500M-1000g** | 74.61 | 44.08 | 50.86 | 43.10 | 30.28 | 30.87 |
| **NT-2500M-multi** | 78.77 | 56.20 | **61.99** | **55.30** | 36.49 | 40.34 |
| **DNABERT-2** | 78.27 | 52.57 | 56.88 | 50.52 | 31.13 | 36.27 |
| **DNABERT-2♦** | **80.17** | **57.42** | 61.90 | 53.00 | **39.89** | **41.20** |

| | Epigenetic Marks Prediction | | | | Promoter Detection | | |
| --- | --- | --- | --- | --- | --- | --- | --- |
| | H3K79me3 | H3K9ac | H4 | H4ac | all | notata | tata |
| **DNABERT (3-mer)** | 60.12 | 50.48 | 78.27 | 38.60 | 90.44 | 93.61 | 69.83 |
| **DNABERT (4-mer)** | 59.77 | 51.44 | 78.28 | 36.40 | 89.54 | 92.65 | 66.78 |
| **DNABERT (5-mer)** | 59.61 | 51.11 | 77.27 | 37.48 | 90.16 | 92.45 | 69.51 |
| **DNABERT (6-mer)** | 61.17 | 51.22 | 79.26 | 37.43 | 90.48 | 93.05 | 61.56 |
| **NT-500M-human** | 58.35 | 45.81 | 76.17 | 33.74 | 87.71 | 90.75 | 78.07 |
| **NT-500M-1000g** | 59.33 | 49.29 | 76.29 | 36.79 | 89.76 | 91.75 | 78.23 |
| **NT-2500M-1000g** | 61.20 | 52.36 | 79.76 | 41.46 | 90.95 | 93.07 | 75.80 |
| **NT-2500M-multi** | 64.70 | 56.01 | 81.67 | 49.13 | **91.01** | 94.00 | **79.43** |
| **DNABERT-2** | **67.39** | 55.63 | 80.71 | **50.43** | 86.77 | 94.27 | 71.59 |
| **DNABERT-2♦** | 65.46 | **57.07** | **81.86** | 50.35 | 88.31 | **94.34** | 68.79 |

| | Transcription Factor Prediction (Human) | | | | | Core Promoter Detection | | |
| --- | --- | --- | --- | --- | --- | --- | --- | --- |
| | 0 | 1 | 2 | 3 | 4 | all | notata | tata |
| **DNABERT (3-mer)** | 67.95 | 70.90 | 60.51 | 53.03 | 69.76 | **70.92** | 69.82 | **78.15** |
| **DNABERT (4-mer)** | 67.90 | 73.05 | 59.52 | 50.37 | 71.23 | 69.00 | 70.04 | 74.25 |
| **DNABERT (5-mer)** | 66.97 | 69.98 | 59.03 | 52.95 | 69.26 | 69.48 | 69.81 | 76.79 |
| **DNABERT (6-mer)** | 66.84 | 70.14 | 61.03 | 51.89 | 70.97 | 68.90 | 70.47 | 76.06 |
| **NT-500M-human** | 61.59 | 66.75 | 53.58 | 42.95 | 60.81 | 63.45 | 64.82 | 71.34 |
| **NT-500M-1000g** | 63.64 | 70.17 | 52.73 | 45.24 | 62.82 | 66.70 | 67.17 | 73.52 |
| **NT-2500M-1000g** | 66.31 | 68.30 | 58.70 | 49.08 | 67.59 | 67.39 | 67.46 | 69.66 |
| **NT-2500M-multi** | 66.64 | 70.28 | 58.72 | 51.65 | 69.34 | 70.33 | **71.58** | 72.97 |
| **DNABERT-2** | **71.99** | **76.06** | **66.52** | **58.54** | **77.43** | 69.37 | 68.04 | 74.17 |
| **DNABERT-2♦** | 69.12 | 71.87 | 62.96 | 55.35 | 74.94 | 67.50 | 69.53 | 76.18 |

| | Transcription Factor Prediction (Mouse) | | | | | Virus | Splice |
| --- | --- | --- | --- | --- | --- | --- | --- |
| | 0 | 1 | 2 | 3 | 4 | Covid | Reconstruct |
| **DNABERT (3-mer)** | 42.31 | 79.10 | 69.90 | 55.40 | 41.97 | 62.23 | 84.14 |
| **DNABERT (4-mer)** | 49.42 | 79.95 | 72.62 | 51.79 | 44.13 | 59.87 | 84.05 |
| **DNABERT (5-mer)** | 42.45 | 79.32 | 62.22 | 49.92 | 40.34 | 50.46 | 84.02 |
| **DNABERT (6-mer)** | 44.42 | 78.94 | 71.44 | 44.89 | 42.48 | 55.50 | 84.07 |
| **NT-500M-human** | 31.04 | 75.04 | 61.67 | 29.17 | 29.27 | 50.82 | 79.71 |
| **NT-500M-1000g** | 39.26 | 75.49 | 64.70 | 33.07 | 34.01 | 52.06 | 80.97 |
| **NT-2500M-1000g** | 48.31 | 80.02 | 70.14 | 42.25 | 43.40 | 66.73 | 85.78 |
| **NT-2500M-multi** | 63.31 | 83.76 | 71.52 | 69.44 | 47.07 | **73.04** | **89.35** |
| **DNABERT-2** | 56.76 | 84.77 | 79.32 | 66.47 | **52.66** | 71.02 | 84.99 |
| **DNABERT-2♦** | **64.23** | **86.28** | **81.28** | **73.49** | 50.80 | 68.49 | 85.93 |

Table 6: This table presents the performance of all the models on the GUE benchmark. ♦: perform further pre-training on the training sets of the GUE benchmark.

|          | EMP | TF-M | CVC | TF-H | PD-tata | PD-o | CPD-tata | CPD-o | SSP |
|----------|-----|------|-----|------|---------|------|----------|-------|-----|
| **Epochs** | 3 | 1k | 8 | 3 | 10 | 4 | 10 | 4 | 5 |

Table 7: The number of training steps we used for the following tasks: Epigenetic Marks Prediction (EMP), Transcription Factor Prediction on the Human genome and the Mouse genome (TF-H and TF-M), Covid Variants Classification (CVC), *tata* dataset of Promoter Detection (PD-tata), *notata* and *all* datasets of Promoter Detection (PD-o), *tata* dataset of Core Promoter Detection (CPD-tata), *notata* and *all* datasets of Core Promoter Detection (CPD-o), and Splice Site Prediction (SSP). In the task of Transcription Factor Prediction on the Mouse genome, we train the model for 1000 steps on each dataset.

Table 6 presents the evaluation results of DNABERT-2 and baselines on the GUE benchmark.

## A.2   IMPLEMENTATION

We pre-train DNABERT-2 with the Masked Language Modeling (MLM) loss with a mask ratio of $15\%$. Notably, we independently mask every token instead of masking spans of continuous tokens like Ji et al. (2021). We use a batch size of 4096 and a max sequence length of 128. We train the model for 500000 steps using the AdamW (Loshchilov & Hutter, 2019) optimizer with $\beta_1 = 0.9$, $\beta_2 = 0.98$, $\epsilon = 1e-6$ and weight decay of $1e-5$. The learning rate linearly increases from 0 to $5e-4$ during the first 30000 steps while linearly decreasing to 0 in the last 470000 steps.

## A.3   HYPERPARAMETERS

This section presents the hyperparameters we used in the fine-tuning stage on each model. Table 7 shows the number of training steps we used for each task. We use **AdamW** (Loshchilov & Hutter, 2019) as optimizer. We keep most of the other hyperparameters the same for all the models across all the datasets, including a batch size of 32, a warmup step of 50, and a weight decay of 0.01. For DNABERT and DNABERT-2, we perform standard fine-tuning with a learning rate of $3e-5$, while for the Nucleotide Transformers, we perform parameter efficient fine-tuning (PEFT) using Low-Rank Adaptation (LoRA) with a learning rate of $1e-4$, a LoRA alpha of 16, a LoRA dropout of 0.05, and a LoRA $r$ of 8. The hyperparameters are selected based on grid searches over commonly used ones in preliminary experiments. The pre-training stage takes approximately 14 days using eight Nvidia RTX 2080Ti GPUs. To train the model, we used the Transformers library (Wolf et al., 2020) and the Composer library (Team, 2021).

## A.4   PRELIMINARY EXPERIMENTS ON NUCLEOTIDE TRANSFORMER

Since there is no official fine-tuning code of Nucleotide Transformer (Dalla-Torre et al., 2023), we use its open-sourced checkpoints in Huggingface Modelhub[2] and train it with our code base using LoRA. For a fair comparison with this model, in this section, we present preliminary experiments that compare the results reported in their paper with the performance of this model under our implementation. We select the epigenetic marks prediction task for benchmarking since it is the only shared task among Dalla-Torre et al. (2023) and GUE. The task contains 10 datasets. For each dataset, we randomly split it into training and test sets with a ratio of 9:1. As shown in Table 8, our LoRA implementation leads to slightly better results than the results reported in the original paper, making our comparison to the model fair and convincing despite the fact that we do not have access to its official fine-tuning implementation.

---

[2]https://huggingface.co/InstaDeepAI

|  | H3 | H3K14ac | H3K36me3 | H3K4me1 | H3K4me2 | H3K4me3 |
|---|---|---|---|---|---|---|
| **500M-human\*** | 72.00 | 37.00 | 45.00 | 36.00 | 27.00 | 24.00 |
| **500M-human** | 69.67 | 33.55 | 44.14 | 37.15 | 30.87 | 24.06 |
| **500M-1000g\*** | 74.00 | 38.00 | 47.00 | 38.00 | 26.00 | 24.00 |
| **500M-1000g** | 72.52 | 39.37 | 45.58 | 40.45 | 31.05 | 26.16 |
| **2500M-1000g\*** | 75.00 | 45.00 | 53.00 | 42.00 | 28.00 | 31.00 |
| **2500M-1000g** | 74.61 | 44.08 | 50.86 | 43.10 | 30.28 | 30.87 |
| **2500M-multi\*** | 79.00 | 54.00 | 62.00 | 54.00 | 32.00 | 41.00 |
| **2500M-multi** | 78.77 | 56.20 | 61.99 | 55.30 | 36.49 | 40.34 |

|  | H3K79me3 | H3K9ac | H4 | H4ac | Average |
|---|---|---|---|---|---|
| **500M-human\*** | 57.00 | 45.00 | 75.00 | 33.00 | 45.10 |
| **500M-human** | 58.35 | 45.81 | 76.17 | 33.74 | **45.35** |
| **500M-1000g\*** | 56.00 | 48.00 | 76.00 | 34.00 | 46.10 |
| **500M-1000g** | 59.33 | 49.29 | 76.29 | 36.79 | **47.68** |
| **2500M-1000g\*** | 57.00 | 49.00 | 79.00 | 41.00 | 50.00 |
| **2500M-1000g** | 61.20 | 52.36 | 79.76 | 41.46 | **50.86** |
| **2500M-multi\*** | 62.00 | 55.00 | 81.00 | 49.00 | 56.90 |
| **2500M-multi** | 64.70 | 56.01 | 81.67 | 49.13 | **58.06** |

Table 8: This table presents the performance of the Nucleotide Transformer on ten datasets of epigenetic marks prediction on the Yeast genome. As shown in the table, our implementation achieves better performance than the results reported in the paper, indicating the fairness of comparison in our experiments. \*: Results taken from Dalla-Torre et al. (2023).

## A.5 Comparing DNABERT-2 with other baselines

We also compare DNABERT-2 with other baselines, including HyenaDNA (Nguyen et al., 2024), a convolutional neural network (CNN) designed for DNA classification by Grešová et al. (2023), and DNABERT-2 model without pre-training. We implement HyenaDNA with their official implementation `https://huggingface.co/LongSafari/hyenadna-medium-450k-seqlen-hf` on HuggingFace and Huggingface Trainer. We implement CNN with official code on GitHub. For both papers, we use the default hyperparameter described in the papers, and we follow our own evaluation setting (e.g., use the same set of hyperparameters across all the datasets and test on the checkpoint with the lowest validation loss).

As shown in Table 9, DNABERT-2 consistently outperforms the baselines. We note that using the official checkpoint of HyenaDNA on Huggingface does not achieve the same level of performance on the EMP task as reported in their paper. This could be due to multiple reasons: 1) the reported results are not based on the open-source checkpoints, 2) the Huggingface trainer does not train the model properly, and 3) more careful hyperparameter searches are needed.

## A.6 Ablation Study on Tokenization Method

In this section, we present an ablation study on the tokenization method: the overlapping K-mer tokenization used in DNABERT v.s. the BPE tokenization used in DNABERT-2. We train two DNABERT-2 variants on the same training data and with the same model architecture and hyperparameters. We use a batch size of 4096, max sequence length of 128, and train each model for $120,000$ steps. Since the vocab size of BPE is 4096, to keep the number of parameters similar enough, we use 6-mer tokenization, which results in 4101 tokens. We then evaluate the two variants on the GUE benchmark. Table 10 presents the results. As shown in the Table, the variant trained with the BPE tokenizer outperforms the K-mer counterparts on 21 out of 28 datasets, with an average score of 65.33 and 60.92, respectively. As a result, we empirically show that BPE leads to better performance than k-mer after being pre-trained and finetuned with the same data, architecture, and hyperparameters,

| | Epigenetic Marks Prediction | | | | | |
| | H3 | H3K14ac | H3K36me3 | H3K4me1 | H3K4me2 | H3K4me3 |
|---|---|---|---|---|---|---|
| **HyenaDNA** | 67.17 | 31.98 | 48.27 | 35.83 | 25.81 | 23.15 |
| **CNN** | 61.52 | 29.73 | 38.60 | 26.06 | 25.76 | 20.52 |
| **DNABERT-2 w/o PT** | 57.36 | 32.25 | 38.07 | 23.80 | 28.57 | 18.12 |
| **DNABERT-2** | **78.27** | **52.57** | **56.88** | **50.52** | **31.13** | **36.27** |

| | Epigenetic Marks Prediction | | | | Promoter Detection | | |
| | H3K79me3 | H3K9ac | H4 | H4ac | all | notata | tata |
|---|---|---|---|---|---|---|---|
| **HyenaDNA** | 54.09 | 50.84 | 73.69 | 38.44 | 47.38 | 52.24 | 5.34 |
| **CNN** | 46.30 | 40.03 | 62.34 | 25.54 | 75.78 | 85.14 | 70.30 |
| **DNABERT-2 w/o PT** | 51.71 | 44.38 | 66.73 | 30.07 | 76.49 | 84.07 | 34.11 |
| **DNABERT-2** | **67.39** | **55.63** | **80.71** | **50.43** | **86.77** | **94.27** | **71.59** |

| | Transcription Factor Prediction (Human) | | | | | Core Promoter Detection | | |
| | 0 | 1 | 2 | 3 | 4 | all | notata | tata |
|---|---|---|---|---|---|---|---|---|
| **HyenaDNA** | 62.3 | 67.86 | 46.85 | 41.78 | 61.23 | 36.95 | 35.38 | 72.87 |
| **CNN** | 53.95 | 63.20 | 45.22 | 29.84 | 61.48 | 58.07 | 60.09 | 69.33 |
| **DNABERT-2 w/o PT** | 59.35 | 59.27 | 46.13 | 31.36 | 60.14 | 54.58 | 58.26 | 47.15 |
| **DNABERT-2** | **84.38** | **87.16** | **83.99** | **79.85** | **87.80** | **69.47** | **68.04** | **74.17** |

| | Transcription Factor Prediction (Mouse) | | | | | Virus | Splice |
| | 0 | 1 | 2 | 3 | 4 | Covid | Reconstruct |
|---|---|---|---|---|---|---|---|
| **HyenaDNA** | 35.62 | 80.50 | 65.34 | 54.20 | 19.17 | 23.27 | 72.67 |
| **CNN** 0 | 31.14 | 59.74 | 63.15 | 45.48 | 27.18 | 22.23 | 76.79 |
| **DNABERT-2 w/o PT** | 30.55 | 63.74 | 59.92 | 24.12 | 27.20 | 69.76 | 46.80 |
| **DNABERT-2** | **81.60** | **92.67** | **92.38** | **84.91** | **76.10** | **71.02** | **84.99** |

Table 9: Benchmark DNABERT-2 with HyenaDNA (Nguyen et al., 2024), CNN (Grešová et al., 2023), and DNABERT-2 without pre-training on GUE.

showing its data efficiency. Also, as shown in Table 6 of the paper, BPE also leads to 3-4 times less computational costs than K-mer.

| Tokenizer | Epigenetic Marks Prediction | | | | | |
|---|---|---|---|---|---|---|
| | **H3** | **H3K14ac** | **H3K36me3** | **H3K4me1** | **H3K4me2** | **H3K4me3** |
| **K-mer** | 74.62 | 42.71 | 47.26 | 39.66 | 25.33 | 27.43 |
| **BPE** | **77.08** | **55.60** | **57.25** | **45.51** | **40.83** | **42.57** |

| Tokenizer | Epigenetic Marks Prediction | | | | Promoter Detection | | |
|---|---|---|---|---|---|---|---|
| | **H3K79me3** | **H3K9ac** | **H4** | **H4ac** | **all** | **notata** | **tata** |
| **K-mer** | 61.03 | 49.35 | 78.61 | 37.14 | 83.78 | **92.65** | 57.75 |
| **BPE** | **66.01** | **56.79** | **80.07** | **54.19** | **85.57** | 92.55 | **60.85** |

| Tokenizer | Transcription Factor Prediction (Human) | | | | | Core Promoter Detection | | |
|---|---|---|---|---|---|---|---|---|
| | **0** | **1** | **2** | **3** | **4** | **all** | **notata** | **tata** |
| **K-mer** | **67.99** | 67.06 | 59.45 | 50.24 | **72.80** | **74.91** | **69.23** | **74.91** |
| **BPE** | 66.99 | **70.98** | **61.40** | **55.10** | 71.31 | 66.28 | 67.99 | 72.73 |

| Tokenizer | Transcription Factor Prediction (Mouse) | | | | | Virus | Splice |
|---|---|---|---|---|---|---|---|
| | **0** | **1** | **2** | **3** | **4** | **Covid** | **Reconstruct** |
| **K-mer** | **48.96** | 81.69 | 81.71 | 63.17 | 42.83 | 62.16 | 77.90 |
| **BPE** | 48.01 | **81.86** | **82.98** | **73.22** | **46.15** | **69.75** | **79.62** |

Table 10: This table presents the ablation study on tokenization methods.

# B  DATA

## B.1  MULTI-SPECIES GENOME FOR PRE-TRAINING

Table 11 lists the 135 species in 7 categories that we randomly selected for genome foundation model pre-training and presents the number of nucleotides we achieved from each species.

| Category | Species | Num. of Nucleotides (M) |
|---|---|---|
| **Fungi** | Ceratobasidium | 655.37 |
| | Claviceps Maximensis | 329.79 |
| | Fusarium Annulatum | 449.98 |
| | Melampsora | 699.52 |
| | Metschnikowia | 109.36 |
| | Mucor Saturninus | 391.17 |
| | Penicillium Chermesinum | 275.81 |
| | Saccharomyces Cerevisiae | 121.54 |
| | Sporopachydermia Quercuum | 155.71 |
| | Tranzscheliella Williamsii | 184.77 |
| | Xylariales | 399.96 |
| **Protozoa** | Phytophthora Sojae | 792.65 |
| | Pythium Apiculatum | 450.99 |
| **Mammalian** | Bubalus Bubalis | 28768.00 |
| | Camelus Dromedarius | 19757.02 |
| | Human | 31372.10 |
| | Macaca Assamensis | 27593.76 |
| | Macaca Nigra | 28217.13 |
| | Mus Musculus | 26545.98 |
| | Peromyscus Californicus | 24677.56 |

*(Continued on next page)*

*(Continued from previous page)*

| Category | Species | Nucleotides (M) |
|---|---|---|
| **Invertebrate** | Brachionus Rubens | 1327.37 |
| | Ceroptres Masudai | 12.95 |
| | Cotesia Typhae | 1866.62 |
| | Croniades Pieria | 3889.85 |
| | Drosophila Athabasca | 1221.16 |
| | Emesis Russula | 4848.08 |
| | Hydra Oligactis | 12597.75 |
| | Meganola Albula | 3604.25 |
| | Oscheius | 383.21 |
| | Rutpela Maculata | 20213.33 |
| **Other Vertebrate** | Anas Zonorhyncha | 11697.08 |
| | Coregonus Clupeaformis | 26824.02 |
| | Gnathonemus Longibarbis | 7314.74 |
| | Myxocyprinus Asiaticus | 23407.19 |
| | Rhipidura Dahli | 10112.96 |
| **Bacteria** | Aeromonas | 47.33 |
| | Agrobacterium | 97.22 |
| | Alcaligenaceae Bacterium | 20.88 |
| | Aliivibrio | 46.48 |
| | Alphaproteobacteria Bacterium | 14.22 |
| | Amycolatopsis Antarctica | 63.43 |
| | Anaerostipes Faecis | 32.00 |
| | Arthrobacter | 36.27 |
| | Atopobium | 28.63 |
| | Bacillus Bc15 | 57.34 |
| | Bacillus Bs3 2021 | 43.51 |
| | Bacterium | 7.54 |
| | Bacteroidetes Bacterium Qs | 8.99 |
| | Breoghania Corrubedonensis | 53.32 |
| | Caldicoprobacter Oshimai | 27.25 |
| | Candidatus Cryptobacteroides Excrementipullorum | 27.63 |
| | Candidatus Dadabacteria Bacterium Rbg Combo | 11.49 |
| | Candidatus Dwaynia Gallinarum | 16.82 |
| | Candidatus Falkowbacteria Bacterium | 13.88 |
| | Candidatus Geothermincola Secundus | 24.76 |
| | Candidatus Gottesmanbacteria Bacterium | 11.08 |
| | Candidatus Nomurabacteria Bacterium Full | 6.29 |
| | Candidatus Portnoybacteria Bacterium Big Fil Rev | 8.17 |
| | Candidatus Regiella Insecticola | 20.62 |
| | Candidatus Roizmanbacteria Bacterium Combo All | 11.13 |
| | Candidatus Rokubacteria Bacterium | 22.06 |
| | Candidatus Saccharibacteria Bacterium | 6.55 |
| | Candidatus Staskawiczbacteria Bacterium Full | 6.79 |
| | Christensenella | 18.75 |
| | Clostridiaceae Bacterium | 29.62 |
| | Clostridiales Bacterium | 16.59 |
| | Clostridium Cag 505 | 21.26 |
| | Clostridium Mcc328 | 36.43 |
| | Clostridium Nexile | 38.43 |
| | Clostridium Uba3521 | 25.99 |
| | Collinsella Urealyticum | 19.45 |
| | Coprobacillus Cateniformis | 38.38 |
| | Cyanobium | 40.33 |
| | Dehalococcoidia Bacterium | 17.59 |

*(Continued on next page)*

*(Continued from previous page)*

| Category | Species | Nucleotides (M) |
|---|---|---|
| | Enterobacteriaceae Bacterium | 41.46 |
| | Evtepia Gabavorous | 24.94 |
| | Firmicutes Bacterium | 36.66 |
| | Fulvivirga | 65.24 |
| | Jeongeupia Chitinilytica | 39.11 |
| | Legionella Endosymbiont Of Polyplax Serrata | 5.30 |
| | Listeria Ilorinensis | 30.31 |
| | Maribacter Cobaltidurans | 46.40 |
| | Marinomonas | 37.73 |
| | Mesorhizobium | 65.15 |
| | Methyloceanibacter Caenitepidi | 34.25 |
| | Microvirga | 68.63 |
| | Mycolicibacter Engbaekii | 45.21 |
| | Novosphingobium | 46.18 |
| | Omnitrophica Wor Bacterium Rbg | 12.52 |
| | Pantoea | 43.14 |
| | Paraburkholderia Edwinii | 82.99 |
| | Parerythrobacter Lutipelagi | 30.98 |
| | Paulownia Witches Phytoplasma | 8.92 |
| | Polaromonas Eurypsychrophila | 41.61 |
| | Prevotella Ag 487 50 53 | 29.63 |
| **Bacteria** | Prevotella Uba3619 | 31.72 |
| | Prevotella Uba634 | 18.51 |
| | Prochlorococcus Ag-321-I09 | 3.29 |
| | Prochlorococcus Ag-363-B18 | 15.54 |
| | Prochlorococcus Ag-402-L19 | 11.17 |
| | Prochlorococcus Scb243 498N4 | 14.12 |
| | Providencia | 41.89 |
| | Pseudomonas 35 E 8 | 63.56 |
| | Pseudomonas Bigb0408 | 59.52 |
| | Pseudomonas P867 | 62.01 |
| | Pseudomonas Promysalinigenes | 50.47 |
| | Roseobacter | 44.14 |
| | Salinicola Peritrichatus | 46.19 |
| | Salmonella S096 02912 | 48.09 |
| | Salmonella Zj-F75 | 47.87 |
| | Sinorhizobium | 65.53 |
| | Sodalis Ligni | 63.85 |
| | Sphaerochaeta | 28.61 |
| | Sphingobacterium | 36.55 |
| | Sphingomonas Carotinifaciens | 37.53 |
| | Sphingomonas Mesophila | 22.91 |
| | Sporosarcina Jiandibaonis | 36.30 |
| | Sporosarcina Ureilytica | 34.37 |
| | Staphylococcus Gdq20D1P | 28.50 |
| | Staphylococcus M0911 | 24.38 |
| | Streptococcus | 22.18 |
| | Streptomyces 8401 | 88.39 |
| | Streptomyces Di166 | 88.71 |
| | Streptomyces Durbertensis | 59.24 |
| | Streptomyces Neau-Yj-81 | 118.84 |
| | Streptomyces Rk74B | 87.36 |
| | Thermopetrobacter | 26.06 |
| | Uncultured Kushneria | 35.31 |
| | Uncultured Phascolarctobacterium | 17.95 |

*(Continued on next page)*

| Category | Species | Nucleotides (M) |
|---|---|---|
| **Bacteria** | Uncultured Proteus | 35.66 |
| | Verrucomicrobiales Bacterium | 3.15 |
| | Vibrio | 41.47 |
| | Victivallis Lenta | 55.45 |
| | Virgibacillus Salexigens | 44.18 |
| | Xanthomonadales Bacterium | 37.47 |

Table 11: Details statistics of the multi-species genome dataset for pre-training.

## B.2 GENOME UNDERSTANDING EVALUATION (GUE & GUE$^+$)

The proposed benchmark Genome Understanding Evaluation (GUE) contains 36 datasets of 9 biological important genome analysis tasks for various different species. To comprehensively evaluate the genome foundation models in modeling variable-length sequences, we select tasks with input lengths ranging from 70 to 10000. Table 12 presents the detailed statistics of each evaluation dataset. The following tasks are included in the GUE benchmark.

**Promoter detection (Human)** focuses on identifying (proximal) promoter regions, crucial sequences in the human genome responsible for instigating transcription. As many primary regulatory elements are located in this region, accurately detecting these sites is instrumental in advancing our grasp of gene regulation mechanisms and pinpointing the genomic underpinnings of numerous diseases. The dataset is divided twofold, TATA and non-TATA, based on whether a TATA box motif is present in the sequence. We extract -249 +50 bp around the transcription start site (TSS) from TATA and non-TATA promoters downloaded from Eukaryotic Promoter Database (EPDnew) (Dreos et al., 2013) and use it as our promoter class. Meanwhile, we construct the non-promoter class with equal-sized randomly selected sequences outside of promoter regions but with TATA motif (TATA non-promoters) or randomly substituted sequences (non-TATA, non-promoters). We also combine the TATA and non-TATA datasets to obtain a combined dataset named *all*.

**Core promoter detection (Human)** is similar to proximal promoter detection with a focus on predicting the core promoter region only, the central region closest to the TSS and start codon. A much shorter context window (center -34 +35 bp around TSS) is provided, making this a more challenging task than proximal promoter prediction.

**Transcription factor binding site prediction (Human)** predicts binding sites of transcription factors (TF), the key proteins that regulate gene expression in the human genome. Their accurate prediction is key to deciphering complex genetic interactions and identifying potential targets for gene therapies. We accessed the legacy 690 ENCODE ChIP-seq experiments (Consortium et al., 2012) via the UCSC genome browser, which encompasses 161 TF binding profiles in 91 human cell lines. We extracted a 101-bp region around the center of each peak as TFBS class and nonoverlapping sequences with the same length and GC content as non-TFBS class. Finally, we randomly select 5 datasets out of a subset of 690 that we curated by heuristically filtering out tasks that are either too trivial (e.g., over 0.95 F1) or too challenging (e.g., less than 0.50 F1) for existing language models.

**Splice site prediction (Human)** predicts splice donor and acceptor sites, which are the exact locations in the human genome where alternative splicing occurs. This prediction is crucial to understanding protein diversity and the implications of aberrant splicing in genetic disorders. The dataset (Wang et al., 2019) consists of 400-bp-long sequences extracted from Ensembl GRCh38 human reference genome. As suggested by Ji et al. (2021), existing models can achieve almost perfect performance on the original dataset, containing 10,000 splice donors, acceptors, and non-splice site sequences, which is overly optimistic on detecting non-canonical sites in reality. As such, we reconstruct the dataset by iteratively adding adversarial examples (unseen false positive predictions in hold-out set) in order to make this task more challenging.

| Task | Metric | Datasets | Train / Dev / Test |
|------|--------|----------|--------------------|
| **Core Promoter Detection** | **mcc** | tata
notata
all | 4904 / 613 / 613
42452 / 5307 / 5307
47356 / 5920 / 5920 |
| **Promoter Detection** | **mcc** | tata
notata
all | 4904 / 613 / 613
42452 / 5307 / 5307
47356 / 5920 / 5920 |
| **Transcription Factor Prediction (Human)** | **mcc** | wgEncodeEH000552
wgEncodeEH000606
wgEncodeEH001546
wgEncodeEH001776
wgEncodeEH002829 | 32378 / 1000 / 1000
30672 / 1000 / 1000
19000 / 1000 / 1000
27294 / 1000 / 1000
19000 / 1000 / 1000 |
| **Splice Site Prediction** | **mcc** | reconstructed | 36496 / 4562 / 4562 |
| **Transcription Factor prediction (Mouse)** | **mcc** | Ch12Nrf2Iggrab
Ch12Znf384hpa004051Iggrab
MelJundIggrab
MelMafkDm2p5dStd
MelNelfeIggrab | 6478 / 810 / 810
53952 / 6745 / 6745
2620 / 328 / 328
1904 / 239 / 239
15064 / 1883 / 1883 |
| **Epigenetic Marks Prediction** | **mcc** | H3
H3K14ac
H3K36me3
H3K4me1
H3K4me2
H3K4me3
H3K79me3
H3K9ac
H4
H4ac | 11971 / 1497 / 1497
26438 / 3305 / 3305
27904 / 3488 / 3488
25341 / 3168 / 3168
24545 / 3069 / 3069
29439 / 3680 / 3680
23069 / 2884 / 2884
22224 / 2779 / 2779
11679 / 1461 / 1461
27275 / 3410 / 3410 |
| **Covid Variant Classification** | **f1** | Covid | 77669 / 7000 / 7000 |
| **Enhancer Promoter Interaction** | **mcc** | GM12878
HeLa-S3
HUVEC
IMR90
K562
NHEK | 10000 / 2000 / 2000
10000 / 2000 / 2000
10000 / 2000 / 2000
10000 / 2000 / 2000
10000 / 2000 / 2000
10000 / 2000 / 2000 |
| **Species Classification** | **mcc** | fungi
virus | 8000 / 1000 / 1000
4000 / 500 / 500 |

Table 12: Statistics of tasks in the GUE benchmark, including the name and the number of training, validation, and test samples in each dataset.

**Enhancer promoter interaction (Human)**    identifies the interactions between enhancers and promoters, integral to regulating gene expression in the human genome. We formulate it as a sequence-pair binary classification problem.

**Species Classification (Virus & Fungi)**    classifies different species based on genome segments. We construct the datasets respectively from the virus and fungi reference genome achieved from GenBank (Benson et al., 2012).

**Transcription factor binding site prediction (Mouse)**    predicts the binding site of transcription factors on mouse genomes. Similar to human binding site data, we obtain mouse ENCODE ChIP-seq data (Stamatoyannopoulos et al., 2012), which is the largest available collection on the UCSC genome browser (n=78). This time, the negative examples are created using dinucleotide shuffling while preserving relative frequencies, while all other settings stay the same as the human TFBS prediction dataset. We also randomly select 5 datasets out of the 78 datasets using the same process described above.

**Epigenetic marks prediction (Yeast)**  predicts epigenetic marks in yeast, modifications on the genetic material that influence gene expression without altering the DNA sequence. Precise prediction of these marks aids in elucidating the role of epigenetics in yeast. We download the 10 datasets from `http://www.jaist.ac.jp/~tran/nucleosome/members.htm` and randomly split each dataset into training, validation, and test sets with a ratio of 8:1:1.

**Covid variant prediction (Virus)**  aims to predict the variant type of the SARS_CoV_2 virus based on 1000-length genome sequences. We download the genomes from the EpiCoV database (Khare et al., 2021) of the Global Initiative on Sharing Avian Influenza Data (GISAID). We consider 9 types of SARS_CoV_2 variants, including *Alpha*, *Beta*, *Delta*, *Eta*, *Gamma*, *Iota*, *Kappa*, *Lambda* and *Zeta*.

