# OpenReview forum: "DNABERT-2: Efficient Foundation Model and Benchmark For Multi-Species Genomes"
_ICLR.cc/2024/Conference — ICLR 2024 poster_

### Official Review · Reviewer_ZSAj · 2023-10-23

**Soundness:** 4 excellent
**Presentation:** 4 excellent
**Contribution:** 2 fair
**Rating:** 6
**Confidence:** 4

**Summary:**

The author proposes a tokenizer for DNA language model, namely, using BPE, in contrast to the k-mer approaches as used before. The authors also propose a large-scale benchmark called GUE to compare DNA language models.

**Strengths:**

- Clear definition of motivations, challenges, and solutions
- The use of BPE makes intuitive sense
- Experiments look solid and extensive
- Solving an important problem of DNA LM
- A new large-scale benchmark

**Weaknesses:**

- Novelty is questionable since BPE is a well-known technique. The use of FlashAttention, LoRA, and AliBi are also not new. So methodologically, it is hard to gauge its novelty.

**Questions:**

- Is there potential for cross-species information leakage? For instance, given the substantial overlap in genomes between humans and primates, the model might easily predict the masked token.

- How does this compare to HyenaDNA?

- On page 7, the authors note that they utilize LoRA for NT but opt for full fine-tuning for DNABERT/DNABERT-2. However, in the methods section, LoRA is described as an integral part of the approach. This is somewhat perplexing.

- While the authors suggest further pre-training on GUE sequences, this might raise concerns regarding its ability to generalize to datasets with novel sequences. For a balanced comparison, it might be best if the authors refrain from additional pre-training on GUE sequences.

- Did the authors evaluate the sequence statistics of the GUE sequences in relation to the sequences from the pre-training corpus?

- The authors claim the method requires significantly less computational power and memory. Did they test the performance with a larger model size? If there wasn't a notable performance enhancement, it would be noteworthy to highlight this.

- Have the authors assessed how the model's performance varies with different dataset sizes?

- Have the authors conducted ablation on FlashAttention, AliBi, and LoRA?

---

> ### Author Response · Authors · 2023-11-16
> **Response to reviewer ZSAj - Part 1**
>
> Thank you for your detailed assessment of our work! We understand your concerns. We have provided much more empirical results to support our claims and provide more detailed explanation to solve the confusion.
>
> **W1: Novelty is questionable since BPE is a well-known technique. The use of FlashAttention, LoRA, and AliBi are also not new. So methodologically, it is hard to gauge its novelty.**
>
> Thank you for your comments regarding the novelty of our work. We understand your concerns about the application of well-known techniques in our study. However, we would like to emphasize that the novelty of our work lies not in the invention of new techniques, but in their innovative application and combination to address specific challenges in genome language modeling, as well as provide solid and well-tailored foundation model and benchmark to this area.
>
> 1. While BPE is established in NLP, our work is among the first to explore the use of BPE for genome sequences. Instead of trivially applying it, we provide a comprehensive theoretical foundation (Section 2) and empirical evidence (Section 1 of our common response) to demonstrate the advantages of BPE in this context. We also conduct a comprehensive empirical analysis of the model’s performance and efficiency with different vocab sizes. We believe our work provides key insights and guidance for future research in this domain.
> 2. The novelty also lies in our unique combination of advanced techniques (FlashAttention, LoRA, and ALiBi) to overcome specific barriers in large genome foundational models. This combination, tailored to the unique challenges, demonstrates how existing tools can be synergistically applied to create a more efficient and effective model for genome analysis. And the pre-trained model itself can serve as an efficient yet powerful tool for countless downstream tasks.
> 3. Additionally, our work introduces the Genome Understanding Evaluation (GUE) benchmark, the largest and most comprehensive benchmark to date for genome foundational models that address the lack of standardized and comprehensive benchmarks in this field.
>
> In summary, while the individual techniques we employed are known, their application in the context of genome language modeling, the way they are combined to address specific challenges, the pre-trained DNABERT-2 model, and the introduction of a comprehensive new benchmark constitute significant novel contributions to our work. We believe these aspects collectively advance the field of genome foundational models and provide valuable insights and resources for future research.
>
> **Q1: Is there potential for cross-species information leakage? For instance, given the substantial overlap in genomes between humans and primates, the model might easily predict the masked token.**
>
> Thank you for indicating this! We agree with the concern of cross-species information leakage. In an extreme case, if all the species have the same genome, than training on the multi-species genome is identical to training the model on a single species genome for lots of epochs, which leads to severe overfitting problems. This is one of the main reasons we chose to train DNABERT-2 on multi-species genomes.
>
> We can also find some evidence in existing works, for example, if we compare the Nucleotide Transformers variants trained on 1000G and multi-species genome, the one trained on the multi-species genome actually performs much better despite the size of the 1000G dataset being larger than the multi-species one. We think the diversity and comprehensiveness of pre-training data is essential to the model’s performance. There may exist more strategical ways of selecting different species to achieve better diversity and comprehensiveness and we will investigate more in this direction in the future.
>
> **Q2: How does this compare to HyenaDNA?**
>
> Thanks for asking. HyenaDNA is a concurrent work of us. We have added it as a baseline and reported its results on the common results. In short, we think DNABERT-2 and HyenaDNA have different focuses. Based on our current evaluation, DNABERT-2 is more effective on different types of tasks, while HyenaDNA, as a convolution-based model, has the potential to handle much longer sequences given the same GPU memory. We will compare with it rigorously and add our comparison results and conclusion in the camera-ready version.

---

> ### Author Response · Authors · 2023-11-16
> **Response to reviewer ZSAj - Part 2**
>
> **Q3: On page 7, the authors note that they utilize LoRA for NT but opt for full fine-tuning for DNABERT/DNABERT-2. However, in the methods section, LoRA is described as an integral part of the approach. This is somewhat perplexing.**
>
> Sorry for causing the confusion. We did make LoRA available in our DNABERT-2 implementation but we did not use it in the final results since it leads to little efficiency improvements given the small size and efficiency of DNABERT-2 and does not affect the performances. We will re-structure the method section in the camera-ready version to avoid confusion. Thanks for pointing out this!
>
> **Q4: While the authors suggest further pre-training on GUE sequences, this might raise concerns regarding its ability to generalize to datasets with novel sequences. For a balanced comparison, it might be best if the authors refrain from additional pre-training on GUE sequences.**
>
> Sorry for causing the confusion! We will explain it better in the camera-ready version. We evaluate two variants of the DNABERT-2 model, **DNABERT-2** and **DNABERT-2♢**, as shown in Tables 2, 3, and 4 in the paper. The first one is the model purely trained on the pre-training corpus (never seen GUE before), while the second one is achieved by further training the first one with MLM loss on GUE's training set. As shown in Table 2, without further pre-training on GUE, **DNABERT-2** already performs on par with **NT-2.5B-multi**. The fact that **DNABERT-2♢** outperforms **DNABERT-2** by 1 on the average score on GUE shows the benefit of further pre-training.
>
> **Q5: Did the authors evaluate the sequence statistics of the GUE sequences in relation to the sequences from the pre-training corpus?**
>
> Thanks for asking! The average sequence length during pre-training is about 700 bp. After adding the 8 datasets (as described in the common response), the sequence length in GUE ranges from 70bp to 10000bp. Based on our evaluations, DNABERT-2 performs consistently well across tasks with different sequence lengths, either the input is 10 times shorter or 15 times longer than the sequences it saw during pre-training. We will add this discussion in the camera-ready version, too.
>
> **Q6: The authors claim the method requires significantly less computational power and memory. Did they test the performance with a larger model size? If there wasn't a notable performance enhancement, it would be noteworthy to highlight this.**
>
> Unfortunately, we do not have enough resources yet to train a much larger model. But we agree with your point on this. The phenomenon you described can imply the existence of some sort of "ceiling" in DNA language models. This is something we are interested in diving into in the future. However, given the fact that a small-scale in-domain pre-training can lead to big performance improvement, we think we are still far from the ceiling of DNA language models.
>
> **Q7: Have the authors assessed how the model's performance varies with different dataset sizes?**
>
> Thanks for pointing out this! We agree it is important, so we design experiments to assess the model on this. We show the results in section 4 of our common response and will add it to the camera-ready version. Thanks!
>
> **Q8: Have the authors conducted ablation on FlashAttention, AliBi, and LoRA?**
>
> Thanks for asking! As shown in Section 1 of the common response, we have done ablation studies on the effect of BPE.
>
> The FlashAttention module, in theory, does not affect the final result at all, since it only changes the process of attention calculation. So we did not perform an ablation study on it. We are performing the ablation study on ALiBi and it may take a few days. We will share the results here is we manage to get it before the end of the discussion period, and we will add it to the camera-ready version, too. As for LoRA, we only perform an ablation study on its effects on Nucleotide Transformers, since NT are the only models we use LoRA. In Appendix A.4 we compare the results of our implementation of NT with LoRA with the results they reported in the paper to show our comparison with them is fair. We do not compare NT on LoRA vs full finetuning, since we do not have enough resources to fully finetune NTs.

---

> ### Author Response · Authors · 2023-11-20
> **A kind remind**
>
> Dear reviewer ZSAj,
>
> Thank you for dedicating your time to reviewing our manuscript. Your insights have been invaluable, and we've crafted a detailed response to address the points you raised. We're confident that our responses have effectively addressed all your concerns. With these changes in mind, we hope you might reconsider and adjust your evaluation of our work.
>
> Should there be any remaining issues, we're more than ready to provide further clarifications. As the rebuttal phase is nearing its deadline, we're looking forward to engaging in a timely discussion. Thank you again for your expertise and guidance!

---

> > ### Comment · Reviewer_ZSAj · 2023-11-22
> >
> > Thanks for the detailed response! I am raising my scores to 6.

---

> > > ### Author Response · Authors · 2023-11-22
> > > **Thank you!**
> > >
> > > Thank you very much! We are happy to know that our responses solve your concerns and delighted to see you raised your score.

---

### Official Review · Reviewer_cdaf · 2023-10-31

**Soundness:** 3 good
**Presentation:** 3 good
**Contribution:** 4 excellent
**Rating:** 8
**Confidence:** 4

**Summary:**

The authors describe a foundation model for DNA sequences, improving on existing models (of which there are relatively few) in terms of computational requirements.

**Strengths:**

-Incorporation of more recent language model techniques into the modeling approach.

- Foundation models have the potential to be a highly useful resource for the computational biology community.  There are very few options at the moment, and the significantly reduced computational requirements of DNABERT2 compared to the Nucleotide Transformer on the one hand, and improved accuracy over DNABERT, make it a welcome addition.

- DNABERT2 is appropriately benchmarked against DNABERT and the Nucleotide Transformer.

- The authors have curated a collection of datasets for benchmarking DNA language models.  The benchmark datasets are sufficiently challenging to provide good discrimination between the performance of the various methods, and indicate that there is still plenty of room for improvement.

**Weaknesses:**

- Deep learning models applied to one-hot encoded genomic sequences appear to have a much higher level of interpretability than those that utilize k-mer tokenization, and I expect this to be even worse for the BPE encoding used in this work.  Unlike other areas of application, in computational biology applications, interpretability is a key factor in choosing a model.

**Questions:**

- "Despite having 30% more parameters than DNABERT, DNABERT-2 requires only one-third the number of FLOPs. This indicates the superiority of the Byte Pair Encoding (BPE)-based tokenization method over overlapping k-mer tokenization in terms of modeling efficiency."
Not sure I agree with this statement - the increased efficiency might be the result of other differences between the models.
"This underscores the importance of providing the model with adequate data, particularly when the model size is scaled up, and further highlights the inefficiency of overlapping k-mer tokenization. The comparison between DNABERT and NT-2500M-1000g exposes the sample inefficiency of non- overlapping k-mer tokenization. Despite being trained on 2.5 times more tokens, NT-2500M-1000g achieves a performance similar to that of DNABERT."
Again, there are other differences between the models, so ascribing this to the difference in tokenization method is a stretch.  If you want to demonstrate the advantage of BPE tokenization, you will need to perform an experiment on two different versions of DNABERT2 - one with k-mer tokenization, and one with BPE tokenization.  **The authors have addressed this point with a thorough ablation study**.

- Please compare your benchmark datasets with the recently published "Genomic benchmarks":
Grešová, K., Martinek, V., Čechák, D. et al. Genomic benchmarks: a collection of datasets for genomic sequence classification. BMC Genom Data 24, 25 (2023). https://doi.org/10.1186/s12863-023-01123-8

typos / grammar:

BENCKMARK: GENOME UNDERSTANDING EVALUATION (GUE)

---

> ### Author Response · Authors · 2023-11-16
> **Response to reviewer cdaf**
>
> Thank you for your encouraging evaluation of our work on DNABERT2. We are especially grateful for your recognition of the model's potential as a valuable resource in the computational biology community, and your acknowledgment of the significant improvements introduced by our model. Following your suggestions, we have added multiple experiments.
>
> **W1 : Deep learning models applied to one-hot encoded genomic sequences appear ot have a much higher level of interpretability than those that utilize k-mer tokenization, and I expect this to be even worse for the BPE encoding used in this work. Unlike other areas of application, in computational biology applications, interpretability is a key factor in choosing a model.**
>
> Thank you for highlighting the critical aspect of interpretability in computational biology applications. We acknowledge that models trained on one-hot encoded sequences offer a high level of interpretability, especially when it comes to understanding the interactions at the level of individual nucleotide bases. In fact, there does exist an inevitable trade-off between fine-grant interpretability and the model’s computational efficiency.
>
> We think BPE tokenization also offers a unique perspective on genomic sequence analysis. BPE tokenization groups frequently co-occurring nucleotide sequences, providing insights into how these 'groups'  play a role in the genome's function. For instance, consider a scenario where a mutation occurs. We may be able to understand how impactful a mutation is by looking at both the difference between the tokenized sequences before and after the mutation, and the model’s predictions/attentions among different tokens.
>
> **Q1: If you want to demonstrate the advantage of BPE tokenization, you will need to perform an experiment on two different versions of DNABERT2 - one with k-mer tokenization, and one with BPE tokenization.**
>
> Thank you for highlighting the importance of directly comparing the effectiveness of different tokenization methods in DNABERT2. We concur that an ablation study focusing on tokenization techniques is crucial for substantiating the superiority of BPE (Byte Pair Encoding) over k-mer tokenization. In response to your suggestion, we have conducted additional experiments with two distinct versions of DNABERT2 - one utilizing k-mer tokenization and the other employing BPE tokenization. As detailed in Section 1 of our common response, these experiments have yielded compelling results. The version of DNABERT2 with BPE tokenization demonstrates a marked improvement in performance over its k-mer counterpart. This outcome not only confirms the efficacy of BPE tokenization in the context of DNA language modeling but also provides robust empirical evidence to support our claim. We believe that these findings significantly bolster the validity of our approach and contribute valuable insights into the optimization of language models for genomic data analysis
>
> **Q2: Please compare your benchmark datasets with the recently published "Genomic benchmarks"**
>
> Thanks for indicating this! In short, GenomicsBenchmark is a concurrent work with an aim similar to us, to create a comprehensive and discriminative space to compare different genome analysis models. It contains 9 datasets from 3 species with input sizes ranging from 200 to ~4000, while GUE contains 36 datasets from multiple species with input sizes ranging from 70 to 1000. Compared to GUE, GenomicsBenchmark also provides a Python package for data loading and processing, which is very helpful and we plan to do this in the future. We will include and discuss this paper in the camera-ready version.

---

> > ### Comment · Reviewer_cdaf · 2023-11-21
> >
> > The authors' rebuttal has addressed my concerns.  Overall the authors were very responsive to reviewer comments, and performed additional experiments that further demonstrate the validity of their approach, and should improve the paper once incorporated.  I expect this will be an impactful paper, and strongly support it being published.  My score already reflected that assessment.

---

> > > ### Author Response · Authors · 2023-11-21
> > > **Thank you**
> > >
> > > Dear reviewer cdaf,
> > >
> > > We are delighted to know that we have solved your concerns. Thank you very much for your recognition and support of this work.

---

> ### Author Response · Authors · 2023-11-20
> **A kind remind**
>
> Dear reviewer cdaf,
>
> Thank you for dedicating your time to reviewing our manuscript, and thanks a lot for your recognition of our novelty and contribution to this field. Your insights have been invaluable, and we've crafted a detailed response to address the points you raised. We're confident that our responses have effectively addressed all your concerns. Should there be any remaining issues, we're more than ready to provide further clarifications. As the rebuttal phase is nearing its deadline, we're looking forward to engaging in a timely discussion. Thank you again for your expertise and guidance!

---

### Official Review · Reviewer_1Kmp · 2023-10-31

**Soundness:** 3 good
**Presentation:** 4 excellent
**Contribution:** 3 good
**Rating:** 6
**Confidence:** 4

**Summary:**

DNABERT2 is an update of the DNABERT, which is an application of the BERT structure to DNA data. My guess is that it first performs tokenisation of input DNA sequence, then pre-trains on DNA dataset to get the token embeddings, after that it adds a few layers to utilise the token embeddings for classification tasks such as promoter detection and transcription factor prediction. The manuscript made the following improvements: (1) use Byte Pair Encoding (BPE) for tokenisation (2) use attention with linear biases (ABiLi) for position encoding and (3) use flash attention and low-rank adaptation (LoRA) for acceleration. It also compiles a larger benchmark dataset for comparing different methods. The manuscript demonstrated that DNABERT2 improved over DNABERT and had a similar performance as Nucleotide transformer.

I think the authors have done a decent amount of work and the work could be more useful for the community if the authors could
(1) perform an ablation study to quantify the contribution of BPE and ALiBi independently.
(2) explain why the code, data and pre-trained model could not be made public now
(3) explain why mcc and f1 are used as the comparison metric for different tasks
(4) explain the benefit of further pre-training. I get lost in understanding the sentence "This results in 0.41B training tokens..." right above section 5.3.

**Strengths:**

DNABERT2 is an update of the DNABERT, which is an application of the BERT structure to DNA data. My guess is that it first performs tokenisation of input DNA sequence, then pre-trains on DNA dataset to get the token embeddings, after that it adds a few layers to utilise the token embeddings for classification tasks such as promoter detection and transcription factor prediction. The manuscript made the following improvements: (1) use Byte Pair Encoding (BPE) for tokenisation (2) use attention with linear biases (ABiLi) for position encoding and (3) use flash attention and low-rank adaptation (LoRA) for acceleration. It also compiles a larger benchmark dataset for comparing different methods. The manuscript demonstrated that DNABERT2 improved over DNABERT and had a similar performance as Nucleotide transformer.

I think the authors have done a decent amount of work and the work could be more useful for the community if the authors could
(1) perform an ablation study to quantify the contribution of BPE and ALiBi independently.
(2) explain why the code, data and pre-trained model could not be made public now
(3) explain why mcc and f1 are used as the comparison metric for different tasks
(4) explain the benefit of further pre-training. I get lost in understanding the sentence "This results in 0.41B training tokens..." right above section 5.3.

**Weaknesses:**

DNABERT2 is an update of the DNABERT, which is an application of the BERT structure to DNA data. My guess is that it first performs tokenisation of input DNA sequence, then pre-trains on DNA dataset to get the token embeddings, after that it adds a few layers to utilise the token embeddings for classification tasks such as promoter detection and transcription factor prediction. The manuscript made the following improvements: (1) use Byte Pair Encoding (BPE) for tokenisation (2) use attention with linear biases (ABiLi) for position encoding and (3) use flash attention and low-rank adaptation (LoRA) for acceleration. It also compiles a larger benchmark dataset for comparing different methods. The manuscript demonstrated that DNABERT2 improved over DNABERT and had a similar performance as Nucleotide transformer.

I think the authors have done a decent amount of work and the work could be more useful for the community if the authors could
(1) perform an ablation study to quantify the contribution of BPE and ALiBi independently.
(2) explain why the code, data and pre-trained model could not be made public now
(3) explain why mcc and f1 are used as the comparison metric for different tasks
(4) explain the benefit of further pre-training. I get lost in understanding the sentence "This results in 0.41B training tokens..." right above section 5.3.

**Questions:**

DNABERT2 is an update of the DNABERT, which is an application of the BERT structure to DNA data. My guess is that it first performs tokenisation of input DNA sequence, then pre-trains on DNA dataset to get the token embeddings, after that it adds a few layers to utilise the token embeddings for classification tasks such as promoter detection and transcription factor prediction. The manuscript made the following improvements: (1) use Byte Pair Encoding (BPE) for tokenisation (2) use attention with linear biases (ABiLi) for position encoding and (3) use flash attention and low-rank adaptation (LoRA) for acceleration. It also compiles a larger benchmark dataset for comparing different methods. The manuscript demonstrated that DNABERT2 improved over DNABERT and had a similar performance as Nucleotide transformer.

I think the authors have done a decent amount of work and the work could be more useful for the community if the authors could
(1) perform an ablation study to quantify the contribution of BPE and ALiBi independently.
(2) explain why the code, data and pre-trained model could not be made public now
(3) explain why mcc and f1 are used as the comparison metric for different tasks
(4) explain the benefit of further pre-training. I get lost in understanding the sentence "This results in 0.41B training tokens..." right above section 5.3.

---

> ### Author Response · Authors · 2023-11-16
> **Response to reviewer 1Kmp**
>
> We are grateful for your positive assessment of DNABERT2 and your acknowledgment of the key improvements we have implemented over the original DNABERT model. We have done more experiments and will polish the paper accordingly to account for them.
>
> **W1: perform an ablation study to quantify the contribution of BPE and ALiBi independently.**
>
> Thanks for indicating this! We totally agree that the ablation study is important for the audience to understand the model. As shown in the common response, we have done an ablation study to compare BPE vs k-mer with the same DNABERT-2 architecture and shown that BPE is indeed more data-efficient. Thank you for this suggestion! We are performing the ablation study on ALiBi and it may take a few days. We will share the results here if we manage to get it before the end of the discussion period, and we will add it to the camera-ready version, too.
>
> **W2 explain why the code, data and pre-trained model could not be made public now**
>
> Sorry for the confusion. To avoid violating the double-blind reviewing process, we follow the common practice to describe our open-sourcing plan in this way. In fact, our code has been publicly available in the supplementary file since the paper submission time. The link to the model and data is also shared in the README file. We will add links to the non-anonymous DNABERT-2 resources in the camera-ready version since we are not allowed to add them now. Thanks for your support to the open-source community.
>
> **W3 explain why mcc and f1 are used as the comparison metric for different tasks**
>
> As suggested by [1], MCC is more reliable than F1 in binary classification and in scenarios when the test set is unbalanced. Since most of our datasets are binary classification and are unbalanced, we use MCC in most cases. The only exception is the virus classification dataset, which is multi-task and perfectly balanced. Thus, we used F1, which we considered to be more common in this scenario. Thanks for indicating the confusion! Since MCC also works well in the multi-class classification case, we plan to always use it as the metric to avoid confusion and will explain it in the camera-ready version.
>
> **W4 explain the benefit of further pre-training. I get lost in understanding the sentence "This results in 0.41B training tokens..." right above section 5.3.**
>
> Sorry for causing the confusion! We will explain it better in the camera-ready version. In general, if we consider there exists a distributional difference between the pre-training corpus and downstream task, further pre-training on the downstream task datasets helps the model to adapt tot the downstream domain. Here we also investigate the effect of further pre-training in the DNA domain.
>
> We evaluate two variants of the DNABERT-2 model, **DNABERT-2** and **DNABERT-2♢**, as shown in Tables 2, 3, and 4 in the paper. The first one is the model purely trained on the pre-training corpus, while the second one is achieved by further training the first one with MLM loss on GUE's training set. The sentence "This results in 0.41B training tokens..." refers to the process of training DNABERT-2 on GUE's training set. Please kindly let me know if this is clearly explained.
>
> As shown in Table 2, **DNABERT-2♢** outperforms **DNABERT-2** by 1 on the average score on GUE, showing the benefit of further pre-training on genome sequence modeling.
>
> [1] Chicco, Davide, and Giuseppe Jurman. "The advantages of the Matthews correlation coefficient (MCC) over F1 score and accuracy in binary classification evaluation." *BMC genomics* 21.1 (2020): 1-13

---

> > ### Comment · Reviewer_1Kmp · 2023-11-19
> > **The authors have addressed all my concerns.**
> >
> > I keep my original rating.

---

> > > ### Author Response · Authors · 2023-11-19
> > > **Thanks for sharing**
> > >
> > > We are happy to know that we have addressed all your concerns. Please don't hesitate to share if you have anything more.

---

### Official Review · Reviewer_cVdN · 2023-11-01

**Soundness:** 3 good
**Presentation:** 3 good
**Contribution:** 2 fair
**Rating:** 6
**Confidence:** 5

**Summary:**

The paper introduces DNABERT-2, an advancement in genome foundation modeling, which aims to decode the linguistic intricacies of genomes. The authors assert that the computational and sample inefficiencies of k-mer tokenization, predominantly used in earlier models, act as barriers in the development of foundational models for large genomes. To address this, the paper introduces Byte Pair Encoding (BPE) as a replacement for k-mer tokenization. BPE is more efficient and overcomes the limitations of the k-mer approach. The authors also emphasize the need for a standardized benchmark for genome understanding and consequently introduce the Genome Understanding Evaluation (GUE) dataset. Experimental results reveal that DNABERT-2 performs on par with state-of-the-art models but with fewer parameters and less GPU time during pre-training. The model also shows significant improvements over the original DNABERT.

**Strengths:**

1. DNABERT-2 incorporates ALiBi and Flashattention mechanisms, enhancing speed and context length.
2. The model successfully borrows several techniques from LLM (Large Language Models) and integrates them into DNABERT.
3. The authors have collected a comprehensive dataset tailored for short sequence prediction.
4. The research is detailed, with a focus on the nuances of the biology setting and the existing benchmarks, showcasing a holistic approach.

**Weaknesses:**

1. The input size for the proposed benchmark seems to be on the shorter side for genomics, potentially limiting its applicability to broader genomics problems.
2. The benchmark's design appears constrained, lacking baseline models like CNNs and omits language model training from scratch, which could provide comparative insights.
3. While the paper is apt for an ML conference, there is a discernible deficiency in the depth of biological insights. Better downstream tasks, such as CAGE-seq prediction and so on.... (longer sequence context)

**Questions:**

1. In the introduction, can you clarify what you specifically mean by "genome language modeling"?
2. Following up on the theme, why was there no citation or reference to models like deepbind/deepSEA? For instance, the TF-DNA binding prediction from Wang et al., 2022, seems not a great citation? Not a genomics language modeling.
3. Given the unique structure and function of DNA, why is there a continued emphasis on tokenization in DNA language modeling?
4. I recommend adding the count of sequences for each dataset in Table 1 to provide a clearer understanding of the dataset sizes.
5. Why weren't tasks involving longer sequences incorporated after introducing DNABERT?

---

> ### Author Response · Authors · 2023-11-16
> **Response to reviewer cVdN (Part 1)**
>
> We sincerely appreciate your thorough evaluation of our work on DNABERT-2 and your insightful comments and questions. Your recognition of the strengths of our research is highly valued. We have added 8 new datasets and 3 baselines to extend GUE’s applicability and provide comparative insights.
>
> **W1: The input size for the proposed benchmark seems to be on the shorter side for genomics, potentially limiting its applicability to broader genomics problems.**
>
> We appreciate your observation regarding the input size limitation in the original version of our Genome Understanding Evaluation (GUE) benchmark. You are correct in noting that the initial input sizes, ranging up to 1000 base pairs (bp), might not fully encompass the broader spectrum of genomics problems that involve longer genomic sequences.
>
> The decision to initially focus on shorter input sizes was driven by practical considerations related to the capabilities of existing models and the computational resources available to us. Specifically, DNABERT's architecture inherently limits it to handling sequences no longer than 512 bp. Similarly, the Nucleotide Transformer is limited for inputs up to 6000 bp, and processing sequences of this length requires substantial memory (45-50GB) on each single GPU, which exceeded our resource capabilities.
>
> However, recognizing the importance of including longer sequences to enhance the comprehensiveness and applicability of GUE, we have now incorporated 8 additional datasets with input sizes ranging from 5000 to 10000 bp. This expansion allows us to better represent the diversity of genomic sequence lengths encountered in real-world genomics problems.
>
> Our updated results with these additional datasets demonstrate DNABERT-2's remarkable capability in handling longer sequences. Despite being trained on sequences up to 700 bp, DNABERT-2 shows impressive performance on sequences between 5k and 10k bp, outperforming the strongest baseline by a significant margin on all 8 of these datasets. This underscores DNABERT-2's robustness and versatility in modeling genomic sequences of varying lengths.
>
> We agree that extra-long input sequences, such as those around 1 million bp, are also very meaningful to study. However, they present unique challenges that might require specifically designed architectures or different approaches. Our current benchmark focuses on sequence lengths that can be effectively handled by existing pre-trained models without necessitating specialized architectural modifications.
>
> **W2: The benchmark's design appears constrained, lacking baseline models like CNNs and omits language model training from scratch, which could provide comparative insights.**
>
> Thank you for your insightful feedback regarding the design of our benchmark and the selection of baseline models. We understand your concern that the initial version of our benchmark may have appeared constrained due to the focus on state-of-the-art (SOTA) models, potentially overlooking the insights that could be gained from including a broader range of baselines.
>
> Recognizing the importance of a more comprehensive comparative analysis, we have expanded our set of baseline models in the revised manuscript. Specifically, we have added two recent CNN models from referenced papers [1] and [2]. These models represent a different approach to genomic sequence analysis and provide a valuable point of comparison to highlight the strengths and potential limitations of our approach.
>
> Furthermore, to address the aspect of language model training from scratch, we have included an additional baseline: DNABERT-2 without pre-training. This allows us to directly assess the impact and effectiveness of pre-training in our model. By comparing DNABERT-2 without pre-training to the pre-trained version, especially when finetuned on the entire dataset versus finetuned on only 5% of the training data, we offer a clearer picture of how pre-training contributes to the model's performance. This comparison is detailed in Section 4 of our common response.
>
> [1] Nguyen, Eric, et al. "Hyenadna: Long-range genomic sequence modeling at single nucleotide resolution." *arXiv preprint arXiv:2306.15794* (2023).
> [2] Grešová, Katarína, et al. "Genomic benchmarks: a collection of datasets for genomic sequence classification." *BMC Genomic Data* 24.1 (2023): 25

---

> ### Author Response · Authors · 2023-11-16
> **Response to reviewer cVdN (Part 2)**
>
> **W3: While the paper is apt for an ML conference, there is a discernible deficiency in the depth of biological insights. Better downstream tasks, such as CAGE-seq prediction and so on.... (longer sequence context)**
>
> We sincerely appreciate your constructive feedback regarding the depth of biological insights in our paper. We acknowledge the importance of incorporating biologically relevant downstream tasks, such as CAGE-seq prediction for transcriptional regulation and activity, to enhance the biological applicability and insights of our model.
>
> However, it is crucial to highlight that the primary goal of this paper was to tackle specific computational challenges in genome foundation models. Our focus was on enhancing the efficiency and effectiveness of these models in handling large and diverse genomic datasets. This approach was driven by a need for more computationally practical yet powerful tools that can manage the increasing scale and complexity of genomic data, a challenge that is pivotal for advancing genome research.
>
> In developing the Genome Understanding Evaluation (GUE) benchmark, our emphasis was slightly different from providing in-depth biological insights. Our aim was to construct a benchmark that, for evaluation purposes, could reveal a clear distinction between the computational capabilities of various methods in genome modeling. To achieve this, we considered several metrics, including the difficulty of tasks, comprehensiveness of datasets, standardization of benchmarks, and applicability to existing genome foundational models. With this focus in mind, tasks and datasets that could best examine and showcase such computational strengths of these models were prioritized.
>
> We recognize the importance of bridging the gap between computational methodology and biological application. In future works, we plan to incorporate more biologically oriented downstream tasks, such as CAGE-seq prediction and other tasks requiring longer sequence contexts and are actively working on it. We believe that incorporating these downstream tasks will not only enrich the biological insights provided by our model but also enhance its practical relevance and utility in real-world genomic research.
>
> **Q1: In the introduction, can you clarify what you specifically mean by "genome language modeling"?**
>
> In our study, the terms 'DNA language modeling' and 'genome language modeling' are used synonymously. By 'genome language modeling', we specifically mean the development of a language model tailored to genomic sequences. This involves leveraging language model techniques to enhance our understanding of the genome, such as predicting genome functionality and interpreting genomic data.
>
> **Q2: Following up on the theme, why was there no citation or reference to models like deepbind/deepSEA? For instance, the TF-DNA binding prediction from Wang et al., 2022, seems not a great citation? Not a genomics language modeling.**
>
> We apologize for the oversight in not citing seminal works such as DeepBind and DeepSEA, which indeed represent significant advancements in the field. To rectify this, we will revise the statement 'Recent advancements in genome language modeling have demonstrated their superiority …' to 'Recent advancements in deep learning techniques for genomic data analysis have demonstrated their superiority ...' in the camera-ready version. This change more accurately reflects the scope of the referenced literature. We will also ensure to include and discuss DeepBind and DeepSEA appropriately, recognizing their critical contributions to the field.

---

> ### Author Response · Authors · 2023-11-16
> **Response to reviewer cVdN (Part 3)**
>
> **Q3: Given the unique structure and function of DNA, why is there a continued emphasis on tokenization in DNA language modeling?**
>
> The concept of tokenization is pivotal in DNA language modeling, mirroring its importance in natural language processing (NLP). In NLP, tokenization involves converting text into smaller units (tokens) for language models to process. Similarly, in DNA language modeling, we tokenize long, continuous sequences of nucleotides into small chunks. This step is crucial as it transforms genomic sequences into a form amenable to language model analysis. The choice of tokenization method significantly influences the model's perception and interpretation of the genomic data. As such, our work delves into various DNA tokenization strategies, assessing their impact and providing valuable insights for future research in this domain.
>
> We have now also demonstrated BPE leads to better model performance than k-mer tokenizations (Ablation study 1 in common response), which may be attributed to the fact that BPE iteratively merges the most frequent co-occurring genome segments that may be biologically relevant. Such observation emphasizes the critical role of proper tokenization on more effectively applying language model techniques to genomic sequences.
>
> **Q4: I recommend adding the count of sequences for each dataset in Table 1 to provide a clearer understanding of the dataset sizes.**
>
> We sincerely appreciate the reviewer’s suggestion. We completely agree and will add the counts to the camera-ready version.
>
> **Q5: Why weren't tasks involving longer sequences incorporated after introducing DNABERT?**
>
> We appreciate the reviewer for raising this important question. We have addressed this in our responses to the weaknesses and have now added 8 additional datasets involving longer sequences.

---

> ### Author Response · Authors · 2023-11-20
> **A kind remind**
>
> Dear reviewer cVdN,
>
> Thank you for dedicating your time to reviewing our manuscript. Your insights have been invaluable, and we've crafted a detailed response to address the points you raised. We're confident that our responses have effectively addressed all your concerns. With these changes in mind, we hope you might reconsider and adjust your evaluation of our work.
>
> Should there be any remaining issues, we're more than ready to provide further clarifications. As the rebuttal phase is nearing its deadline, we're looking forward to engaging in a timely discussion. Thank you again for your expertise and guidance!

---

> > ### Comment · Reviewer_cVdN · 2023-11-20
> > **Thanks for the response**
> >
> > I'm very happy and satisfied with the response to Weakness 2 and Questions 1, 2, 4.
> >
> > However, for Weakness 1, I believe HyenaDNA also includes species classification tasks in their paper(and even much longer input size). Why not include them for comparison? And for Q3, I believe the base-pair is a more nature option. Regarding other tasks such as CAGE-seq prediction, I am still not convinced about why they should not be incorporated. I understand that most papers on DNA language models, like NT transformer and HyenaDNA, don't cover this, but I think it's a very important issue in this field.
> >
> > Overall, I really appreciate the response and the additional experiments, so I have raised my score from 5 to 6.

---

> > > ### Author Response · Authors · 2023-11-20
> > > **Thanks for raising your score and for the follow-up questions**
> > >
> > > Dear reviewer cVdN,
> > >
> > > Thanks for sharing your opinions! We are delighted that we solved most of your concerns and you raised your evaluation of our work. We are more than glad to help with other concerns.
> > >
> > > 1. Thanks for indicating this. We did not choose to use the dataset HyenaDNA used for multiple reasons.
> > >     - The dataset is not standardized. There is no clear split of train/dev/test, which does not follow our criterion of constructing the benchmark.
> > >     - The dataset is relatively easy. For example, HyenaDNA achieves an accuracy of over 0.93 with an input size of 32k. The fungi and virus datasets we provided contain more species and therefore more appropriate in terms of difficulty when evaluating different models.
> > >      - The input size of that dataset (up to 1M) is out of range for almost all the existing models except HyenaDNA thanks to its CNN-based architecture. Based on our evaluation, with a Transformer architecture and appropriate optimization (e.g., mixed precision) on 80GB A100 GPUs, DNABERT-2 can handle up to 115k inputs while 2.5B NT models can only handle up to 10k inputs, which are far away from 1M.
> > >
> > > Due to the above reasons, we designed the long sequence datasets with 5k-10k inputs and clearly defined train/dev/test samples to ensure it is applicable to most of the existing models and appropriately discriminate the capability of different models. Nevertheless, we will try to train the models on this dataset to compare with the HyenaDNA model.
> > >
> > >
> > > 2. We agree that the base pair is a more natural option for tokenization. However, other tokenization methods offer their unique benefits. For example, they reduce the input sequence length by 5-6 times, which is especially helpful when Transformer-based architecture is used. Without those specific tokenization mechanisms, on an 80GB A100 GPU, the max input length of DNABERT-2 will reduce from 115k to 23k, and that of 2.5B NT will reduce from 10k to 1.5k. The greatly reduced max input length can greatly restrict the applicability of the models. It is possible to perform specific model architecture design to account for the huge input size, but a well-studied tokenization method can also orthogonally serve as a type of information extraction and compression.
> > >
> > > 3. We agree that tasks such as CAGE-seq prediction are important and we are in the way of constructing a core promoter prediction dataset based on the CAGE-seq data. We are finetuning different models on it, modifying the setup, and trying to understand the results statistically. Yet we haven't finished all the processes rigorously. We may also add it into GUE in the camera-ready version if we manage to rigorously standardize this task. Thanks for this suggestions.

---

> > > > ### Author Response · Authors · 2023-11-21
> > > > **Add species classification results on the dataset HyenaDNA used**
> > > >
> > > > Dear reviewer cVdN,
> > > >
> > > > Thanks for your suggestion to include the species classification dataset used by HyenaDNA. We have constructed the 5-species classification dataset described in the HyenaDNA paper, where train/dev/test split is achieved across different chromosomes. Following their official GitHub repo, we build a dataset with 1k/1k/1k samples respectively for train/dev/test with an input size of 32k. We finetune DNABERT-2 and baselines on this dataset to show the effectiveness of DNABERT-2 in handling extra-long input.
> > > >
> > > > | seq_len | 1k | 32k | 250k | 450k |
> > > > | --- | --- | --- |  --- |  --- |
> > > > | HyenaDNA | $\underline{61.10}$ |  $\underline{93.40}$ |  $\underline{97.90}$ |  $\underline{99.40}$ |
> > > > | DNABERT | 39.50 | 92.00 | N/A | N/A |
> > > > | NT 2.5B | 58.10 | N/A | N/A | N/A |
> > > > | DNABERT-2 | 61.00 | 99.30 | N/A | N/A |
> > > >
> > > > Please note that our results are not directly comparable with HyenaDNA's, since our data split is not exactly the same. However, using it as a reference, we show the effectiveness of DNABERT-2 in handling extra-long input. With 32k inputs, it achieves a similar level of performance as HyenaDNA using 450k input. We will also include this dataset in GUE to account for extra-long input modeling.

---

> > > > > ### Comment · Reviewer_cVdN · 2023-11-23
> > > > > **Thanks for the response**
> > > > >
> > > > > Considering the benchmark for long-length tasks and tokenization, I will maintain the score at 6

---

### Author Response · Authors · 2023-11-16
**Common Response to all the reviewers - Part 1**

We sincerely thank all the reviewers for providing comprehensive and in-depth reviews that greatly help to improve the quality and presentation of the current manuscript. We appreciate the reviewers' recognition of our presentation, contributions, and novelty. We will incorporate all the results we showed here in the camera-ready version.

Following your great suggestions, we provide additional empirical results from several perspectives to solve concerns and confusion in our first manuscript:

1. **Ablation Study on Tokenization Method: BPE v.s. K-mer**,
2. **Expansion of GUE with 8 Additional Datasets with 5k-10k inputs**,
3. **Inclusion of 3 New Baselines**.
4. **Performance Analysis with Varied Data Portions** (5%, 10%, 20%, 50%, 100%),

We would love to share the results we have at this moment. We acknowledge that due to time and resource constraints, some of the experiments recommended by the reviewers are still ongoing. We are committed to continuing these experiments and will share the results as soon as they are available.

**Please note, when we refer to DNABERT-2, we always refer to the one that is purely trained on the pre-training corpus, instead of the one that has been further pre-trained on GUE.**


## **1. Ablation Study on Tokenization Method: BPE v.s. K-mer**

We perform ablation studies on the impact of tokenization methods: BPE v.s.  k-mer. We want to answer the following question: given the same data, architecture, and pre-training cost (GPU hours), which tokenization method leads to better performance?

To fairly compare different tokenization methods, we use the same DNABERT-2 architecture and hyperparameters to pre-train two models from scratch for 120k steps with a batch size of 4096. Since the vocab size of BPE is 4096, to keep the number of parameters similar enough, we use 6-mer tokenization, which results in 4101 tokens (In Tables 2, 3, and 4, we showed that 3-, 4-, 5-, and 6-mers result in very similar performance). We then evaluate these two models on the GUE benchmark. We also use the same hyperparameter in finetuning. We report the average results of 2 runs.

Overall, BPE outperforms K-mer on 21 out of 28 datasets, with an average score of 65.33 and 60.92, respectively. The detailed results of each task are shown as follows:


|       | H3        | H3K14ac  | H3K36me3  | H3K4me1   | H3K4me2   | H3K4me3   | H3K79me3  | H3K9ac    | H4        | H4ac      |
| ----- | --------- | -------- | --------- | --------- | --------- | --------- | --------- | --------- | --------- | --------- |
| BPE   | **77.08** | **55.6** | **57.25** | **45.51** | **40.83** | **42.57** | **66.01** | **56.79** | **80.07** | **54.19** |
| K-mer | 74.62     | 42.71    | 47.26     | 39.66     | 25.33     | 27.43     | 61.03     | 49.35     | 78.61     | 37.14     |



|       | tf-h-0    | tf-h-1    | tf-h-2   | tf-h-3   | tf-h-4   | tf-m-0    | tf-m-1    | tf-m-2    | tf-m-3    | tf-m-4    |
| ----- | --------- | --------- | -------- | -------- | -------- | --------- | --------- | --------- | --------- | --------- |
| BPE   | 66.99     | **70.98** | **61.4** | **55.1** | 71.31    | 48.01     | **81.86** | **82.98** | **73.22** | **46.15** |
| K-mer | **67.99** | 67.06     | 59.45    | 50.24    | **72.8** | **48.96** | 81.69     | 81.71     | 63.17     | 42.83     |



|       | core-tata | core-notata | core-all  | 300-tata  | 300-notata | 300-all   | splice    | covid_variants |
| ----- | --------- | ----------- | --------- | --------- | ---------- | --------- | --------- | -------------- |
| BPE   | 72.73     | 67.99       | 66.28     | **60.85** | 92.55      | **85.57** | **79.62** | **69.75**      |
| K-mer | **74.91** | **69.23**   | **68.45** | 57.75     | **92.65**  | 83.78     | 77.9      | 62.16          |



As a result, we empirically show that BPE leads to better performance than k-mer after being pre-trained and finetuned with the same data, architecture, and hyperparameters, showing its data efficiency. Also, as shown in Table 2 of the paper, BPE also leads to 3-4 times less computational costs than k-mer.

---

### Author Response · Authors · 2023-11-16
**Common Response to all the reviewers - Part 2**

## **2. Expansion of GUE with 8 Additional Datasets**

As identified by reviewer **cVdN**, the current version of the GUE benchmark focuses more on datasets with relatively short input sizes (from 70 to 1000). Since the absence of datasets with longer inputs may prevent us from fully illustrating the capability of the DNABERT-2 model and limits GUE's applicability, we decided to add 8 long sequence datasets. The first 2 datasets focus on species classification, with the first one aiming to classify fungi, and the second one aiming to classify viruses. The rest 6 datasets focus on enhancer-promoter interaction (EPI) prediction, which is formally a sequence-pair classification problem. This addition is particularly significant as it introduces a more complex and challenging task into our benchmark suite, testing our model's ability to understand and predict interactions between different genomic regions. The updated GUE dataset is not available [here](https://filebin.net/sv780vlpw8x3sc2k) (anonymously).

The dataset is described as follows:

| Dataset | Task | seq_len | num_classes | train/dev/test |
| --- | --- | --- | --- | --- |
| GM12878 | enhancer-promoter interaction | 5000 | 2 | 10000/2000/2000 |
| HeLa-S3 | enhancer-promoter interaction | 5000 | 2 | 10000/2000/2000 |
| HUVEC | enhancer-promoter interaction | 5000 | 2 | 10000/2000/2000 |
| IMR90 | enhancer-promoter interaction | 5000 | 2 | 10000/2000/2000 |
| NHEK | enhancer-promoter interaction | 5000 | 2 | 10000/2000/2000 |
| K562 | enhancer-promoter interaction | 5000 | 2 | 10000/2000/2000 |
| virus | species classification | 5000 | 25 | 4000/500/500 |
| fungi | species classification | 10000 | 20 | 8000/1000/1000 |

Now the GUE benchmark consists of 36 datasets of 9 tasks (both single-sequence and sequence-pair tasks), with input size ranging from 70 - 10000, which is the largest and most comprehensive genome classification benchmark to the best of our knowledge. Here we present the performance of DNABERT-2, DNABERT_6mer, and NT-2.5B-multispecies on these tasks. The results of all the models will be reported in the camera-ready version. We report the average results of 2 runs.

(For DNABERT, due to its 512 input limitation, we follow its origin setting by splitting the input into 512-len pieces, generating embedding independently, and averaging the embedding. We are not able to fit it on the EPI task after trying multiple sets of hyperparameters.)


|               | GM12878   | HeLa-S3   | HUVEC    | IMR90     | NHEK     | K562      | virus    | fungi     |
| ------------- | --------- | --------- | -------- | --------- | -------- | --------- | -------- | --------- |
| DNABERT       | -         | -         | -        | -         | -        | -         | 44.51    | 89.29     |
| NT-2.5B-multi | 61.91     | 72.15     | 73.13    | 79.49     | 86.48    | 68.64     | 45       | 92.85     |
| DNABERT-2     | **76.21** | **79.19** | **83.5** | **86.71** | **92.9** | **73.73** | **48.5** | **93.04** |


As shown in the table, DNABERT-2 outperform baselines by a large margin on all the tasks, illustrating DNABERT-2 superiority in handling long sequence. We further compare DNABERT-2 and DNABERT_6mer on variable input sequence lengths ranging from 500 to 10000 on viral and fungi species classification datasets.

Fungi:

| seq_len | 500 | 1000 | 2000 | 5000 | 10000 |
| --- | --- | --- | --- | --- | --- |
| DNABERT | 41 | 55.16 | 71.24 | 83.62 | 89.29 |
| DNABERT-2 | 41.28 | 59.32 | 74.9 | 88.13 | 93.04 |

Virus:

| seq_len | 500 | 1000 | 2000 | 5000 |
| --- | --- | --- | --- | --- |
| DNABERT | 22.03 | 31.27 | 35.94 | 44.51 |
| DNABERT-2 | 23.69 | 31.95 | 37.64 | 48.5 |

As shown in the tables, the performance gap between DNABERT and DNABERT-2 keeps increasing as the sequence length becomes larger. It is worth noting that, the average length of sequences used to pre-train DNABERT-2 is about 700bp, yet it performs greatly on sequences with 10000bp inputs, indicating DNABERT-2's extrapolation capability provided by ALiBi.

---

### Author Response · Authors · 2023-11-16
**Common Response to all the reviewers - Part 3**

## **3. Inclusion of 3 New Baselines**.

As suggested by reviewer **cVdN**, **cdaf**, and **ZSAj**, we add three baseline models to provide comparative insights: HyenaDNA [1], a CNN from GenomicBenchmarks [2], and DNABERT-2 without pre-training. We report the average results of 2 runs.

|  | H3 | H3K14ac | H3K36me3 | H3K4me1 | H3K4me2 | H3K4me3 | H3K79me3 | H3K9ac | H4 | H4ac |
| --- | --- | --- | --- | --- | --- | --- | --- | --- | --- | --- |
| HyenaDNA | 67.17 | 31.98 | 48.27 | 35.83 | 25.81 | 23.15 | 54.09 | 50.84 | 73.69 | 38.44 |
| CNN | 61.52 | 29.73 | 38.6 | 26.06 | 25.76 | 20.52 | 46.3 | 40.03 | 62.34 | 25.54 |
| ours w/o pretrain | 57.36 | 32.25 | 38.07 | 23.8 | 28.57 | 18.12 | 51.71 | 44.38 | 66.73 | 30.07 |
| ours w/ pretrain | 78.27 | 52.57 | 56.88 | 50.52 | 31.13 | 36.27 | 67.39 | 55.63 | 80.71 | 50.43 |

|  | tf-h-0 | tf-h-1 | tf-h-2 | tf-h-3 | tf-h-4 | tf-m-0 | tf-m-1 | tf-m-2 | tf-m-3 | tf-m-4 |
| --- | --- | --- | --- | --- | --- | --- | --- | --- | --- | --- |
| HyenaDNA | 62.34 | 67.86 | 46.85 | 41.78 | 61.2 | 35.62 | 80.5 | 65.34 | 54.2 | 19.17 |
| CNN | 53.95 | 63.2 | 45.22 | 29.84 | 61.48 | 31.14 | 59.74 | 63.15 | 45.48 | 27.18 |
| ours w/o pretrain | 59.35 | 59.27 | 46.13 | 31.36 | 60.14 | 30.55 | 63.74 | 59.92 | 24.12 | 27.2 |
| ours w/ pretrain | 84.38 | 87.16 | 83.99 | 79.85 | 87.8 | 81.6 | 92.67 | 92.38 | 84.91 | 76.1 |

|  | core-tata | core-notata | core-all | 300-tata | 300-notata | 300-all | splice | covid_variants |
| --- | --- | --- | --- | --- | --- | --- | --- | --- |
| HyenaDNA | 72.87 | 35.38 | 36.95 | 5.34 | 52.24 | 47.38 | 2.67 | 23.27 |
| CNN | 69.33 | 60.09 | 58.07 | 70.3 | 85.14 | 75.78 | 76.79 | 22.23 |
| ours w/o pretrain | 47.15 | 58.26 | 54.58 | 34.11 | 84.07 | 76.49 | 46.8 | 69.76 |
| ours w/ pretrain | 74.17 | 68.04 | 69.37 | 71.59 | 94.27 | 86.77 | 84.99 | 71.02 |

We implement HyenaDNA with their [official implementation on HuggingFace](https://huggingface.co/LongSafari/hyenadna-medium-450k-seqlen-hf) and Huggingface Trainer. We implement CNN with [official code on GitHub](https://github.com/ML-Bioinfo-CEITEC/genomic_benchmarks/blob/main/src/genomic_benchmarks/models/torch.py). For both papers, we use the default hyperparameter described in the papers and we follow our own evaluation setting (e.g., use the same set of hyperparameters across all the datasets, and test on the checkpoint with the lowest validation loss). We note that using the official implementation on Huggingface does not achieve the same level of performance on the EMP task as reported in their paper. This could be due to multiple reasons: 1) the reported results are not based on the open-source implementation, 2) the Huggingface trainer does not train the model properly, and 3)  more careful hyperparameter searches are needed. However, at this moment, we are not able to spend more effort and resources on this and will leave this to the future. We share [our scripts](https://file.io/F5CyNlDpDEGf) used in training HyenaDNA if you would love to double-check them.

As shown in the results, DNABERT-2 achieves great improvement through pre-training and outperforms the CNN model by a large margin.

---

### Author Response · Authors · 2023-11-16
**Common Response to all the reviewers - Part 4**

## **4. Performance Analysis with Varied Data Portions**

As suggested by reviewer **ZSAj**, we investigate the impact of dataset size on DNABERT-2. To achieves this, we downsampled 4 variant of the GUE benchmark, with the same test set and respectively 5%, 10%, 20%, and 50% of training and validation data. We also compare with the DNABERT-2 model without pre-training but utilize all the data. We report the average results of 3 runs.

| % of data | H3 | H3K14ac | H3K36me3 | H3K4me1 | H3K4me2 | H3K4me3 | H3K79me3 | H3K9ac | H4 | H4ac |
| --- | --- | --- | --- | --- | --- | --- | --- | --- | --- | --- |
| 100 w/o pretrain | 57.36 | 32.25 | 38.07 | 23.8 | 28.57 | 18.12 | 51.71 | 44.38 | 66.73 | 30.07 |
| 5 | 60.83 | 35.39 | 43.81 | 34.04 | 17.99 | 22.69 | 56.96 | 43.67 | 74.23 | 29.22 |
| 10 | 66.26 | 38.2 | 45.93 | 37.22 | 23.79 | 23.79 | 59.21 | 44.83 | 75.83 | 33.23 |
| 20 | 71.24 | 41.07 | 49.65 | 41.02 | 26.17 | 28.04 | 60.69 | 48.16 | 77.32 | 37.5 |
| 50 | 75.39 | 46.5 | 54.9 | 45.97 | 19.46 | 30.57 | 61.25 | 51.78 | 80.28 | 44.5 |
| 100 | 78.27 | 52.57 | 56.88 | 50.52 | 31.13 | 36.27 | 67.39 | 55.63 | 80.71 | 50.43 |

| % of data | core-tata | core-notata | core-all | 300-tata | 300-notata | 300-all | splice |
| --- | --- | --- | --- | --- | --- | --- | --- |
| 100 w/o pretrain | 47.15 | 58.26 | 54.58 | 34.11 | 84.07 | 76.49 | 46.8 |
| 5 | 32.28 | 63.95 | 57.46 | 22.58 | 90.06 | 81.24 | 38.79 |
| 10 | 43.3 | 64.11 | 58.29 | 44.87 | 91.77 | 81.61 | 74.1 |
| 20 | 58.16 | 66.19 | 61.28 | 53.58 | 91.44 | 82.77 | 78.27 |
| 50 | 65.75 | 67.32 | 61.59 | 53.4 | 93.36 | 86.13 | 81 |
| 100 | 74.17 | 68.04 | 69.37 | 71.59 | 94.27 | 86.77 | 84.99 |

| % of data | tf-h-0 | tf-h-1 | tf-h-2 | tf-h-3 | tf-h-4 | tf-m-0 | tf-m-1 | tf-m-2 | tf-m-3 | tf-m-4 |
| --- | --- | --- | --- | --- | --- | --- | --- | --- | --- | --- |
| 100 w/o pretrain | 59.35 | 59.27 | 46.13 | 31.36 | 60.14 | 30.55 | 63.74 | 59.92 | 24.12 | 27.2 |
| 5 | 63.81 | 69.03 | 46.76 | 42.39 | 60.73 | 25.67 | 80.04 | 64.75 | 26.98 | 28.16 |
| 10 | 63.48 | 67.26 | 50.26 | 42.34 | 61.87 | 36.25 | 81.09 | 66.74 | 27.46 | 43.83 |
| 20 | 67.36 | 71.19 | 60.02 | 44.18 | 70.41 | 46.77 | 84.06 | 73.85 | 38.5 | 43.86 |
| 50 | 67.31 | 70.9 | 63.04 | 47.58 | 73.78 | 54.62 | 85.59 | 75 | 68.16 | 48.8 |
| 100 | 84.38 | 87.16 | 83.99 | 79.85 | 87.8 | 81.6 | 92.67 | 92.38 | 84.91 | 76.1 |

As shown in the Tables, DNABERT-2's performance consistently improves as the size of training/validation data increase. It also shows that DNABERT-2 achieves good performances in most of the datasets when only 5 percent of data is presented. The pre-trained DNABERT-2 outperforms the non-pre-trained one in most scenarios, with  5 percent of training data, showing the great impact of pre-training.

[1] Nguyen, Eric, et al. "Hyenadna: Long-range genomic sequence modeling at single nucleotide resolution." *arXiv preprint arXiv:2306.15794* (2023).

[2] Grešová, Katarína, et al. "Genomic benchmarks: a collection of datasets for genomic sequence classification." *BMC Genomic Data* 24.1 (2023): 25

[3] Ji, Yanrong, et al. "DNABERT: pre-trained Bidirectional Encoder Representations from Transformers model for DNA-language in genome." *Bioinformatics* 37.15 (2021): 2112-2120.

---

### Author Response · Authors · 2023-11-18
**A kind remind of our response**

Dear reviewers,

Thank you very much for providing these valuable and in-depth reviews, and sharing your confusion and concerns. We have added various experiments and provided more details explanations of the terms and setups in the paper. Hope our responses solve all your concerns and confusion. Please kindly let us know if you have any further questions or suggestions. We are more than glad to solve them.

---

### Author Response · Authors · 2023-11-21
**Adding the species classification dataset used in HyenaDNA**

Dear reviewers,

As suggested by reviewer cVdN, we have constructed the 5-species classification dataset described in the HyenaDNA paper, where train/dev/test split is achieved across different chromosomes. Following their official GitHub repo, we build a dataset with 1k/1k/1k samples respectively for train/dev/test with an input size of 32k. We finetune DNABERT-2 and baselines on this dataset to show the effectiveness of DNABERT-2 in handling extra-long input.


| seq_len | 1k | 32k | 250k | 450k |
| --- | --- | --- |  --- |  --- |
| HyenaDNA | $\underline{61.10}$ |  $\underline{93.40}$ |  $\underline{97.90}$ |  $\underline{99.40}$ |
| DNABERT | 39.50 | 92.00 | N/A | N/A |
| NT 2.5B | 58.10 | N/A | N/A | N/A |
| DNABERT-2 | 61.00 | 99.30 | N/A | N/A |

Please note that our results are not directly comparable with HyenaDNA's, since our data split is not exactly the same. However, using it as a reference, we show the effectiveness of DNABERT-2 in handling extra-long input. With 32k inputs, it achieves a similar level of performance as HyenaDNA using 450k input. We will also include this dataset in GUE to account for extra-long input modeling.

---

### Meta-Review · Area_Chair_sPkw · 2023-12-05

**Metareview:**

This paper improves upon DNABERT by replacing k-mer tokenization with byte pair encoding. The paper also adds several improvements that have come about in the transformer encoder architecture. The authors also make available a multi-species data set for benchmarking. They evaluated the model using promoter prediction and enhancer prediction tasks and show an improvement over DNABERT and another model while using many fewer parameters. The reviewers were impressed with the improvements and during discussion, the authors addressed the concerns raised. One concern was that the novelty seemed to be low, but the authors responded clearly that "the novelty of our work lies not in the invention of new techniques, but in their innovative application and combination to address specific challenges in genome language modeling, as well as provide solid and well-tailored foundation model and benchmark to this area." I think this clearly and succinctly states the novelty and contribution. The engineering accomplishments of this work is impressive and the benefits to the community and the scientific goals is clear.

**Justification For Why Not Higher Score:**

The paper reviews do not seem to justify a higher score.

**Justification For Why Not Lower Score:**

The reviewers were nearly uniformly supportive of acceptance. There were concerns about novelty, but I think the authors have clearly placed their work in the context in which the novelty is clear and valued.

---

### Decision · Program_Chairs · 2024-01-16

Accept (poster)